# Valence and spin states of iron are invisible in Earth's lower mantle

Jiachao Liu [1], Susannah M. Dorfman [1], Feng Zhu [2], Jie Li [2], Yonggang Wang [3], Dongzhou Zhang [4], Yuming Xiao[5], Wenli Bi[6,7] & E. Ercan Alp[6]

Heterogeneity in Earth's mantle is a record of chemical and dynamic processes over Earth's history. The geophysical signatures of heterogeneity can only be interpreted with quantitative constraints on effects of major elements such as iron on physical properties including density, compressibility, and electrical conductivity. However, deconvolution of the effects of multiple valence and spin states of iron in bridgmanite (Bdg), the most abundant mineral in the lower mantle, has been challenging. Here we show through a study of a ferric-iron-only $(Mg_{0.46}Fe^{3+}_{0.53})(Si_{0.49}Fe^{3+}_{0.51})O_3$ Bdg that $Fe^{3+}$ in the octahedral site undergoes a spin transition between 43 and 53 GPa at 300 K. The resolved effects of the spin transition on density, bulk sound velocity, and electrical conductivity are smaller than previous estimations, consistent with the smooth depth profiles from geophysical observations. For likely mantle compositions, the valence state of iron has minor effects on density and sound velocities relative to major cation composition.

[1] Department of Earth and Environmental Sciences, Michigan State University, East Lansing, MI 48824, USA. [2] Department of Earth and Environmental Sciences, University of Michigan, Ann Arbor, MI 48109, USA. [3] Center for High Pressure Science and Technology Advanced Research (HPSTAR), Beijing 100094, China. [4] School of Ocean and Earth Science and Technology, Hawai'i Institute of Geophysics and Planetology, University of Hawaii at Manoa, Honolulu, HI 966822, USA. [5] HPCAT, Geophysical Laboratory, Carnegie Institution of Washington, Argonne, IL 60439, USA. [6] Advanced Photon Source, Argonne National Laboratory, Argonne, IL 60439, USA. [7] Department of Geology, University of Illinois at Urbana-Champaign, Urbana, IL 61801, USA. Correspondence and requests for materials should be addressed to J.L. (email: jiacliu09@gmail.com) or to S.M.D. (email: dorfman3@msu.edu)

Variation of redox conditions in the mantle, both laterally and vertically, is a natural consequence of differentiation and mixing processes in the mantle over its history. Early Earth processes segregated reduced iron through the mantle to the metallic core and generated the habitable oxygen-rich surface. Modern convection carries oxidized and iron-rich subducted basalt to the lower mantle[1], while plumes ascending from the lowermost mantle may be reduced[2]. Regional and depth variation of oxygen fugacity, $fO_2$, in the mantle has been confirmed by compositional variability in basalts[3] and mantle mineral inclusions in diamonds[4,5].

Constraints on mantle redox heterogeneity are also important to interpreting remote observations of heterogeneous geophysical properties. Geophysical methods, such as seismic[6–8], geoid[9,10] and geomagnetic[11,12] observations, have been applied globally to image thermochemical variability in the mantle. Besides subducted slabs, observed heterogeneity in seismic tomography includes large-scale features at the base of the lower mantle, which may be thermochemical piles. Two large low shear velocity provinces (LLSVPs) located nearly antipodally beneath the Pacific Ocean and Africa are characterized by lower-than-average shear ($V_S$) and compressional ($V_p$) wave velocities[6,7], and possibly elevated bulk sound velocity ($V_B$)[13] and density[14,15]. Evidence for chemical differences between these regions and the surrounding mantle includes sharp margins and anti-correlated anomalies between $V_B$ and $V_S$ in LLSVPs[16]. Although the identity and origin of these regions is still unknown, the likely high density of LLSVPs may be explained by enrichment in Fe[17].

Redox heterogeneity is likely to be expressed by differences in $Fe^{3+}/\Sigma Fe$ in mantle minerals, but the effects of $Fe^{3+}/\Sigma Fe$ ratios on observable mantle properties and the corresponding influence on the geophysical and geochemical evolution of the Earth are not well understood. The lower mantle's dominant mineral (Mg, Fe, Al)(Fe, Al, Si)$O_3$ bridgmanite (Bdg) accommodates both $Fe^{2+}$ and $Fe^{3+}$, with each species corresponding to potentially different effects on thermoelastic and transport properties[18]. The effects of $Fe^{2+}$ and $Fe^{3+}$ on incompressibility of Bdg are thought to be opposite[19]. The density contrast between $Fe^{2+}$- and $Fe^{3+}$-dominant Bdg may result in separation of oxidized and reduced materials through mantle convection and leave imprints in geochemical and isotopic compositions[20]. However, in many experimental studies on Bdg, $Fe^{3+}/\Sigma Fe$ was not characterized. Moreover, the compositions of Bdg synthesized in laser heated diamond anvil cells (DACs) are, in general, not well-controlled due to unknown oxygen fugacity, inhomogeneity in micron-scale starting materials, and cation migration by Soret diffusion at high temperatures. Such uncertainties in chemistry hamper the investigation of the effects of $Fe^{3+}/\Sigma Fe$ on thermoelastic and electrical properties of Bdg.

Pressure-driven electronic spin-pairing transitions of iron could further distinguish oxidized from reduced Bdg. High-pressure experimental and theoretical studies have concluded that $Fe^{3+}$ in the octahedral B-site of Bdg undergoes a high spin (HS) to low spin (LS) transition under lower mantle pressure–temperature ($P–T$) conditions (e.g., refs. [21–26]). Although this spin transition is generally accepted, discrepancies remain in the pressure conditions of the transition reported in previous experimental studies, e.g., 18–25 GPa[24] vs. 50–70 GPa[21,22]. These differences could originate from experimental protocol (e.g., ref. [24]) or composition-dependence of the spin transition (e.g., ref.[26]). In contrast to $Fe^{3+}$ in the B-site, both $Fe^{2+}$ and $Fe^{3+}$ accommodated in the larger pseudo-dodecahedral A-site will not experience a spin transition under the mantle $P–T$ conditions (reviewed by ref. [27]), though some authors have suggested a transition of $Fe^{2+}$ to an intermediate-spin state[28,29], which has not been supported by theoretical calculations[23,30]. As

a result, the spin transition is only likely to influence the thermoelastic and transport properties of Bdg with $Fe^{3+}$ in the B-site. Geophysical relevance of spin transitions in mantle minerals has been debated, as throughout most of the lower mantle, properties such as seismic wave speeds[31] and electrical conductivity[11,12] do not exhibit discontinuous changes with depth. On the other hand, the spin transition in ferropericlase (Fp) has been suggested to generate a viscosity minimum around 1600 km with important implications for mantle dynamics and interpretation of the geoid[32,33]. If a spin transition in Bdg occurs at similar depths, it may have similar effects on viscosity. Constraints on the effects of the spin transition in Bdg on density, elasticity, viscosity, and thermal and electrical conductivities are key to resolving the geophysical behavior of oxidized regions of the lower mantle.

To disentangle valence and spin effects on the elastic and electrical behavior of Bdg under high pressures, we conducted X-ray diffraction (XRD), X-ray emission spectroscopy (XES), time-domain synchrotron Mössbauer spectroscopy (SMS) and electrical conductivity measurements on $(Mg_{0.46}Fe^{3+}_{0.53})(Si_{0.49}Fe^{3+}_{0.51})O_3$ Bdg at lower mantle pressures up to 85 GPa and 300 K. These complementary results from our well-characterized Bdg sample demonstrate that the spin transition of $Fe^{3+}$ in the Bdg B-site happens between 43 and 53 GPa at 300 K. With improved constraints on the effects of $Fe^{3+}$ on the equation of state (EoS) and electrical conductivity of Bdg, we conclude that neither oxidation state nor spin state of Fe in Bdg would cause significant anomalies in geophysical properties of mantle heterogeneities.

## Results

**Synthesis and characterization of Bdg.** A unique opportunity to unambiguously determine the behavior of oxidized, Al-free Bdg at lower mantle conditions was presented by our discovery of a complete, reversible phase transition at 22–26 GPa and 300 K from $Fe^{3+}$-bearing akimotoite to Bdg. A representative full-profile Le Bail refinement of Bdg at 44.8 GPa is shown in Fig. 1, where all peaks were identified as orthorhombic GdFeO$_3$-type Bdg, Au, or Ne. Purely ferric Bdg with $Fe^{3+}$ evenly distributed between the A- and B-sites is ideal for studying the spin transition of $Fe^{3+}$ because variations of its density, spin moment, hyperfine parameters, and electrical conductivity with respect to pressure are not influenced by $Fe^{2+}$ or cation exchange. The composition

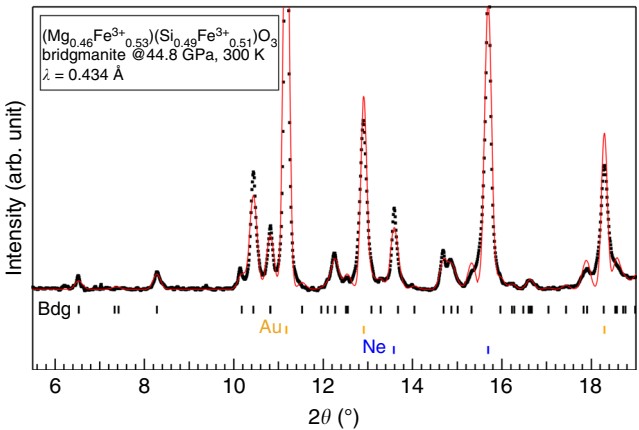

**Fig. 1** Full-profile Le Bail refinement confirms transformation of akimotoite to bridgmanite (Bdg) structure. Measured XRD data for $(Mg_{0.46}Fe^{3+}_{0.53})(Si_{0.49}Fe^{3+}_{0.51})O_3$ composition at 44.8 GPa and 300 K (black dots) are consistent with orthorhombic GdFeO$_3$-type Bdg (black ticks below). Le Bail fit (red curve) also includes expected peak positions for Au calibrant (yellow ticks) and Ne medium (blue ticks)

of the akimotoite starting material was determined by electron microprobe analysis to be $Mg_{0.46}Fe_{1.04}Si_{0.49}O_3$ (Supplementary Fig. 1). In Bdg synthesized from this composition at 26–71 GPa, SMS (Supplementary Fig. 2) are composed of two Fe sites of equal weight with quadrupole splitting (QS) values $< 1.5$ mm s$^{-1}$ and similar center shift (CS) values ($\Delta CS < 0.3$ mm s$^{-1}$, Supplementary Table 1). These values are consistent with the hyperfine parameters for $Fe^{3+}$ of Bdg derived from synchrotron-based energy-domain Mössbauer spectroscopy[34,35] (Supplementary Fig. 3). This confirms that all iron in the Bdg sample is $Fe^{3+}$ and stoichiometry suggests that $Fe^{3+}$ is distributed almost evenly between the A- and B-sites, yielding a Bdg formula of $(Mg_{0.46}Fe^{3+}_{0.53})(Si_{0.49}Fe^{3+}_{0.51})O_3$.

**Spin transition of ferric iron in Bdg**. Complementary XRD and XES results show that the spin transition of $Fe^{3+}$ in the B-site occurs between 43 and 53 GPa at 300 K in $(Mg_{0.46}Fe^{3+}_{0.53})(Si_{0.49}Fe^{3+}_{0.51})O_3$ Bdg (Figs. 2 and 3). Over this pressure range, the compressibility of this Bdg increases sharply and the unit cell volume decreases by about 1.9% (Fig. 2 and Supplementary Table 2). This softening is clear in the decrease in the normalized stress $F$ (Fig. 2, inset), which is sensitive to magnetic and spin transitions under pressure[36]. At pressures below 43 GPa and above 53 GPa, the slope of $F$ vs. Eulerian strain $f$ is almost 0, indicating that the pressure derivative of bulk modulus ($K'$) is nearly 4 and a second order Birch–Murnaghan EoS suffices for fitting these two segments (Fig. 2). Relative to HS $Fe^{3+}$-bearing Bdg, LS Bdg exhibits 2.7% smaller ambient-pressure volume, $V_0$,

and 5.7% higher ambient-pressure bulk modulus, $K_0$ (Supplementary Table 3). The spin transition in our Bdg is confirmed by XES measurements up to 85 GPa at 300 K (Fig. 3a). A total spin moment decreases from a maximum of 2.5, corresponding to 100% HS $Fe^{3+}$, to a minimum of about 1.5 (Fig. 3b), corresponding to 50% HS, 50% LS $Fe^{3+}$, over the range 40–60 GPa (Fig. 3b).

The observed spin transition pressure and volume collapse provide robust confirmation for recent density functional theory calculations and resolve disagreement among previous experimental studies. Theoretical computation[26] found a spin transition in B-site $Fe^{3+}$ at 48–56 GPa and 0 K for a similar composition $(Mg_{0.5}Fe^{3+}_{0.5})(Si_{0.5}Fe^{3+}_{0.5})O_3$. For this composition, no prediction of the effect of the spin transition on the EoS is available, but for a less-enriched $(Mg_{0.875}Fe_{0.125})(Si_{0.875}Fe_{0.125})O_3$ Bdg the spin transition was predicted to result in a volume collapse of 0.5%[37] or 1.2%[23]. The lower bound predicted for $\Delta V$ is consistent with our observations (Fig. 2), assuming a linear relation between $\Delta V$ and iron content. In comparison, Mao et al.[24] reported a 0.5% reduction in unit cell volume at 18–25 GPa with 0.02 $Fe^{3+}$ per formula unit, which is higher but comparable with theoretical prediction[26]. Theoretical calculations predict that the spin transition in A-site $Fe^{3+}$ happens at much higher pressures than the transition in the B-site[23]; therefore, the 50% LS $Fe^{3+}$ derived from our XES data is consistent with the transition of only B-site $Fe^{3+}$ to the LS state at the lower mantle pressures. Previous experimental studies disagreed on the spin transition pressure range: a subtle change in the EoS was reported in a recent study at 18–25 GPa[24], while two other studies found less obvious discontinuities in bulk modulus around 50–70 GPa[21,22]. Other experimental studies observed no spin transition at all (e.g., refs. [38,39]). Differences between observed spin transition pressures are unlikely to be explained by compositional differences alone as had been suggested by computational work[26]: our sample exhibits a spin transition pressure in-between reported pressures in previous experiments on Bdg but has the highest $Fe^{3+}$ content. Different experimental protocols and possible diffusion or reduction of iron during high-temperature experiments could cause the discrepancy. Well-characterized Bdg samples synthesized in the multi-anvil apparatus often incorporate all Fe in the A-site (e.g., refs. [24,38]), and would not be expected to undergo spin transitions under the mantle pressures. Many other studies do not have strong constraints on the valence state or site occupancy of Fe in Bdg, but it is likely that failure to observe spin transitions indicates that no $Fe^{3+}$ is present in the B-site. Moreover, some Bdg samples synthesized using laser heated DACs exhibit excess $SiO_2$, indicating that the composition of synthesized Bdg differs from the starting material. Upon heating, cations may also be oxidized or reduced and/or migrate between the two crystallographic sites[40], and thus some apparent changes in compressibility may be due to different crystal chemistry. Our EoS and XES data obtained on well-characterized samples without any heating during compression provide support for theoretical predictions[23,25,26,37] and experimental observations[21,22] that at lower mantle pressures, A-site $Fe^{3+}$ remains in HS state and B-site $Fe^{3+}$ undergoes the HS–LS transition.

For iron-rich compositions, the elastic properties and spin-transition-induced softening in $Fe^{3+}$-Bdg can be easily distinguished from elastic properties of $Fe^{2+}$-dominant Bdg, but for mantle-relevant amounts of iron this difference becomes insignificant (Fig. 4). With the highest Fe content among synthesized Bdg, our $Fe^{3+}$-only Bdg has the largest unit cell observed to date for Bdg below the pressures of the spin transition (Supplementary Fig. 4). Above the spin transition pressures of B-site $Fe^{3+}$, the unit cell volume of our $Fe^{3+}$-Bdg collapses to match volumes of $Fe^{2+}$-dominant Bdg with similar total Fe content (Supplementary

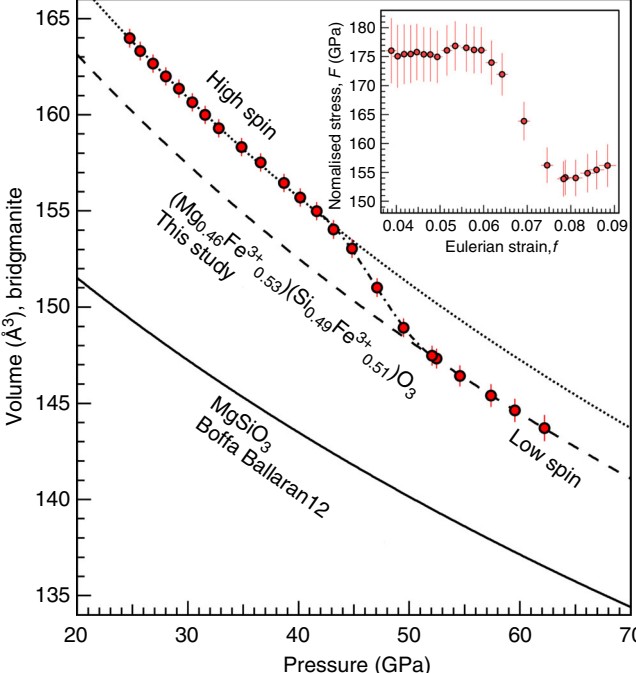

**Fig. 2** Compression behavior of bridgmanite (Bdg) at 300 K across the spin transition. An unit cell volume of $(Mg_{0.46}Fe^{3+}_{0.53})(Si_{0.49}Fe^{3+}_{0.51})O_3$ Bdg and second order Birch–Murnaghan equation of state fits to the high-spin data between 24.7 and 43.1 GPa (dotted) and 50% low-spin data between 52.5 and 61.4 GPa (dashed). Softening is observed between 43.1 and 52.5 GPa (dot dashed). The compression curve of $MgSiO_3$ Bdg is also plotted for comparison (black curve[38]). Inset: normalized stress $F$ vs. Eulerian strain $f$ calculated using the fitted 1-bar unit cell volume from the lower pressure segment, revealing a discontinuity between 43.1 and 52.5 GPa. The error bars are 95% confidence intervals

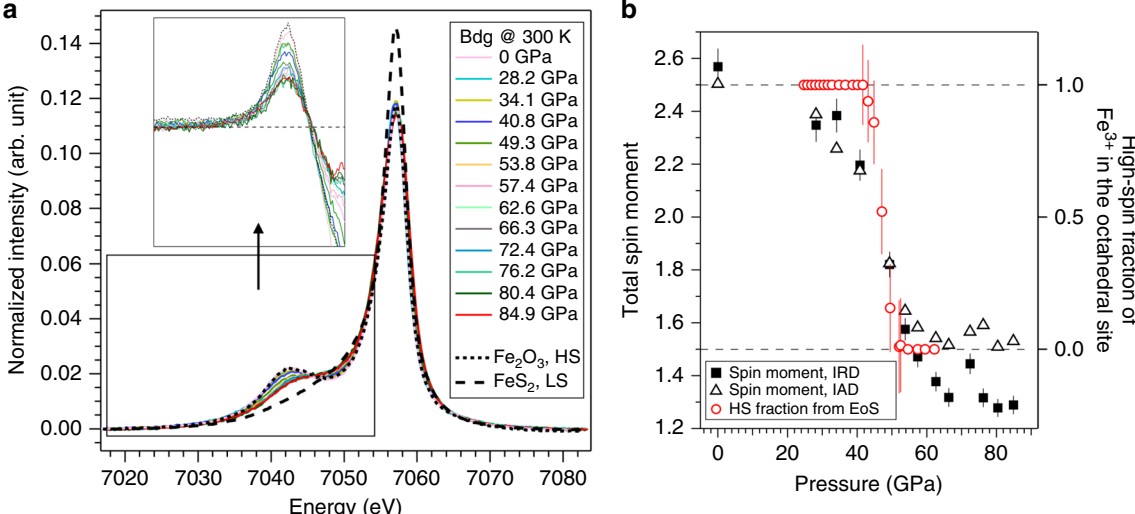

**Fig. 3** X-ray emission Fe $K_\beta$ spectra and total spin moment. **a** XES data for $(Mg_{0.46}Fe^{3+}_{0.53})(Si_{0.49}Fe^{3+}_{0.51})O_3$ bridgmanite up to 84.9 GPa at 300 K. All spectra were normalized to area and aligned to position of the main peak. Both the spectra of $Fe_2O_3$ and the sample at 1 bar served as the high-spin (HS) reference, while $FeS_2$ at 1 bar was used as the low-spin (LS) reference. The inset shows the difference between the sample spectra and the LS reference $FeS_2$. **b** The total spin moment (left axis) derived from both integrated absolute difference (IAD, open triangles) and integrated relative difference (IRD, black squares) methods[68] as a function of pressure. Error bars were determined by evaluating the difference in calculated spin moment using either $Fe_2O_3$ or ambient $Mg_{0.46}Fe^{3+}_{1.04}Si_{0.49}O_3$ sample as the HS standard. The expected spin moment for HS-only configuration is 2.5 (marked by the upper horizontal dashed line), while the B-site spin transition should lower total spin moment to 1.5 (marked by the lower horizontal dashed line). The red open circles are HS fractions of B-site $Fe^{3+}$ (right axis) derived from equation of state (EoS) under high pressures. Error bars of HS fractions obtained from EoS represent 95% confidence interval in EoS parameters $V_{HS}$, $V_{LS}$, $K_{HS}$, and $K_{LS}$[32]. XES and XRD concur that spin transition of B-site $Fe^{3+}$ is centered at 48–49 GPa at ambient temperature

Fig. 4). Consequently, redox heterogeneity cannot be determined from density heterogeneity once the spin transition of B-site $Fe^{3+}$ is complete in the deep lower mantle (Fig. 4a and Supplementary Fig. 5). The bulk moduli $K$ of both HS and LS $Fe^{3+}$-rich Bdg are lower than that of $Fe^{2+}$-dominant Bdg (Fig. 4b). At a representative mid-lower mantle pressure of 80 GPa (corresponding to a depth of 1850 km), $K$ of HS $(Mg_{0.46}Fe^{3+}_{0.53})(Si_{0.49}Fe^{3+}_{0.51})O_3$ Bdg is 11.1% lower than the extrapolated $K$ for $FeSiO_3$ Bdg, and $K$ of $(Mg_{0.46}Fe^{3+}_{0.53})(Si_{0.49}Fe^{3+}_{0.51})O_3$ Bdg with B-site LS $Fe^{3+}$ is 9.3% lower than that of $FeSiO_3$ Bdg (Fig. 4b). The magnitudes of these differences in $K$ are comparable to softening caused by A-site vacancy[41]. The corresponding bulk sound velocity for $Fe^{3+}$-dominant Bdg exhibits a similar trend as bulk modulus (Fig. 4c). The heterogeneity parameter $\partial \ln V_B / \partial X_{Fe}$ for $Fe^{3+}$-Bdg is 0.15; this is 1.5 times of the 0.1 obtained for $Fe^{2+}$-dominant Bdg[19], resulting in a stronger velocity anomaly for an oxidized mantle heterogeneity. If interpolated to a typical mantle composition with iron content $2Fe/(Mg + Fe + Al + Si)$ ~0.1 in Bdg[42], differences in density, bulk modulus, and bulk sound velocity between reduced and oxidized Bdg at 80 GPa are up to 0.3%, 1.1%, and 0.5%, respectively. These small differences have been within experimental uncertainties for studies with less Fe, but can be resolved by our study of well-characterized Fe-rich Bdg samples with careful high-pressure experimental design. Given the fact that lower mantle temperatures would reduce the difference in density and sound velocity between $Fe^{2+}$- and $Fe^{3+}$-bearing bdg, reduced and oxidized Bdg with mantle-relevant iron content will exhibit almost identical seismic velocities in the deep lower mantle.

For a given concentration of Fe, the presence of Al in Bdg has been observed to have relatively minor effects on density and bulk modulus[19,25] (Fig. 4) and may suppress the spin transition by occupying the B-site(see Implications below). As a result,

experiments on Fe, Al-bearing compositions have been unable to unambiguously determine whether and under what conditions spin transitions take place in the mantle. The effects of spin and valence states of Fe on density and bulk compressibility are expected to be even less significant in Al-bearing lithologies in the mantle. Although shear properties cannot be constrained by our experimental data, theoretical calculations have predicted that the effects of trivalent cations and/or spin transition of the B-site $Fe^{3+}$ on shear modulus are even smaller than on bulk modulus[25]. Therefore, the incorporation of trivalent cations in Bdg is not expected to cause obvious elastic anomalies in the lower mantle.

An independent constraint on mantle compositional and thermal heterogeneities can be obtained from lower mantle electrical conductivity. Current electrical conductivity models based on geomagnetic observations show a smooth profile of electrical conductivity with depth in the lower mantle[11,12]. This profile appears to be inconsistent with spin transitions of iron in lower mantle minerals because such a transition reduces the number of unpaired electrons, resulting in a decrease in the mobility and density of the electric charge carriers and a potentially observable decrease in electrical conductivity. The decrease in conductivity due to the spin transition has been observed in Fp[43,44], but has been unclear for Bdg[34,45,46]. Ohta et al.[45] reported a ~0.5 order of magnitude decrease in electrical conductivity at 70–85 GPa in $(Mg_{0.9}Fe_{0.1})SiO_3$ Bdg and attributed this anomaly to the spin transition of $Fe^{3+}$, but two more recent studies reported monotonic increase in electrical conductivity of Bdg under the lower mantle pressures[34,46] (Fig. 5), which are more consistent with electrical conductivity models[11,12]. In order to clarify the influence of spin transition on the electrical conductivity of Bdg, we determined the electrical conductivity of our Bdg sample by using a four-point-probe method (Supplementary Fig. 6). Note that this method is only applicable to Bdg compositions, which can be either recovered or synthesized

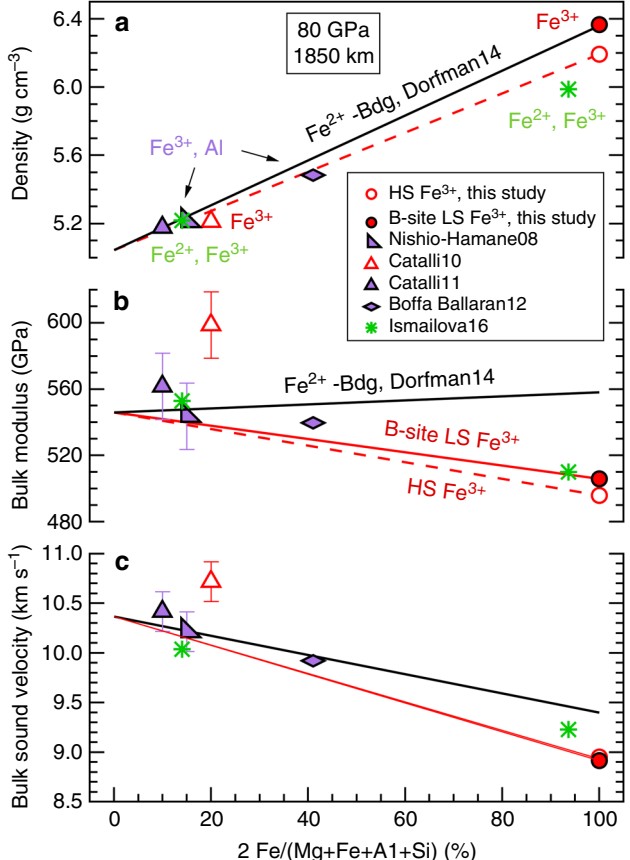

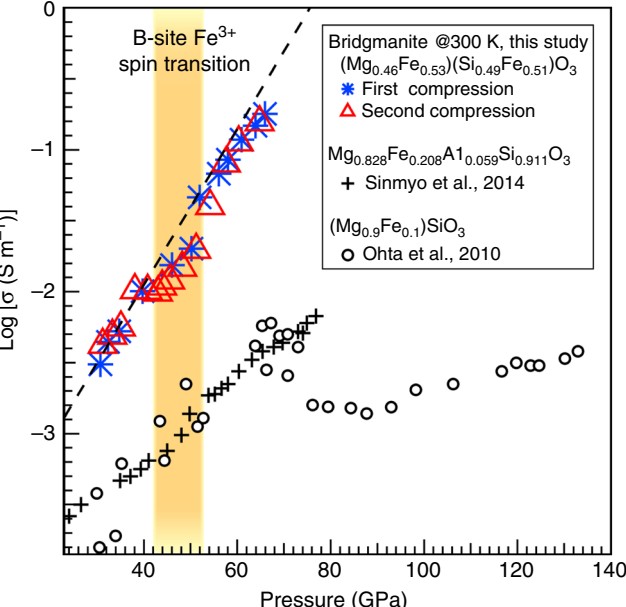

**Fig. 4** Variation of observable seismic properties of bridgmanite as a function of iron content at 80 GPa and 300 K. EoS results from this study and previous work summarized in ref. [19] demonstrate that **a** density **b** bulk modulus, and **c** bulk sound velocity exhibit different dependence on $Fe^{2+}$ and $Fe^{3+}$ content. The red solid lines are linear interpolations between $MgSiO_3$-Bdg and end members for high-spin (HS) $Fe^{3+}$-Bdg and the red dashed lines are those for B-site low-spin (LS) $Fe^{3+}$ Bdg. The black lines are linear fits for $Fe^{2+}$-Bdg summarized in ref. [19]. The $Fe^{3+}$-Bdg end member is from this study, and the open and solid circles are for HS and B-site LS $Fe^{3+}$ Bdg, respectively. Differences between [21] (red triangles) and solid red line trend for bulk modulus and sound velocity may be caused by compositional changes during Bdg synthesis from glass in the laser heated diamond anvil cell. The purple symbols are for $Fe^{3+}$, Al-bearing Bdg samples [22,70], and the green symbols are for $Fe^{3+}$, $Fe^{2+}$-bearing Bdg samples with $Fe^{3+}/\Sigma Fe$ <50% [41]

**Fig. 5** The electrical conductivity of three bridgmanite compositions across the spin transition at high pressures and 300 K. The red open triangles and blue asterisks are for $(Mg_{0.46}Fe^{3+}_{0.53})(Si_{0.49}Fe^{3+}_{0.51})O_3$ Bdg along two successive compression paths by using the same DAC. The uncertainty is smaller than the symbol size. The yellow region marks the pressure range (43–53 GPa) of the spin transition in the B-site $Fe^{3+}$, in this study constrained by complementary XRD and XES measurements. The dashed line is a linear fit to the electrical conductivity data up to 40 GPa, which predicts slightly higher conductivity than measured data above 54 GPa. Between 40 and 54 GPa, there is a 0.18–0.29 log unit drop in electrical conductivity as a result of spin transition in the B-site $Fe^{3+}$. In comparison, Ohta et al. [45] reported a more significant drop (~0.5 log unit) between 70 and 85 GPa in $(Mg_{0.9}Fe_{0.1})SiO_3$ Bdg (black open circles). In contrast to ref. [45], ref. [46] reported much smoother conductivity profile for $Mg_{0.828}Fe_{0.208}Al_{0.059}Si_{0.911}O_3$ Bdg. Note that the conductivity trend from ref. [46] also exhibits a dip within the spin transition pressure range from this study

without laser heating, as Au probes must be attached at ambient conditions to homogeneous samples. The 300-K akimotoite-Bdg transition provides an entirely new route to access electrical properties of $Fe^{3+}$-bearing Bdg. Our results show that the pressure range of spin transition in B-site $Fe^{3+}$ coincides with a subtle decrease of 0.18–0.29 log unit in electrical conductivity (Fig. 5), and this decrease in conductivity was reproduced in two successive experiments using the same DAC. On the other hand, the electrical conductivity of B-site LS $Fe^{3+}$ Bdg is only slightly lower than extrapolated values from the HS segment (Fig. 5), revealing much lower reduction of electrical conductivity by spin transition in Bdg than Fp [43,44]. Given the fact that Fe content in the lower mantle is about one tenth of that in our sample (e.g., ref. [42]) and mantle temperatures would further weaken or broaden the effects of the spin transition, our results demonstrate that the spin transition of B-site $Fe^{3+}$ of Bdg in the lower mantle has a negligible effect on electrical conductivity of the mantle, which is

consistent with the smooth profile obtained from geophysical observations [11,12].

## Discussion

Whether a spin transition occurs in Bdg in Earth's mantle has been subject to debate due both to observed smooth variation in geophysical properties and uncertainty in the crystal chemistry of Fe in Bdg. The $Fe^{3+}/\Sigma Fe$ ratio of Bdg in the lower mantle has been estimated based on sound velocity of Bdg obtained by experimental [47] and theoretical methods [48] to range from 60% to even 90%. This high $Fe^{3+}$ content relative to the upper mantle could be explained by $Fe^{2+}$ self-disproportionation to $Fe^{3+}$ and metallic Fe during the formation of Bdg beneath the transition zone [49]. $Al^{3+}$ facilitates $Fe^{3+}$-enrichment in lower mantle Bdg through the coupled-substitution mechanism ($Mg^{2+}_A + Si^{4+}_B = Fe^{3+}_A + (Fe^{3+}, Al^{3+})_B$) (e.g., refs. [49,50]). Whether $Fe^{3+}$ enters the B-site of Bdg through this coupled-substitution mechanism and further undergoes the spin transition in the lower mantle depends on the concentration of cations available to fill the B-site of Bdg and *P–T* conditions. For Bdg samples synthesized from pyrolitic starting materials representing a lower mantle lithology, observed $Al/Fe^{3+}$ ratios are consistently greater than 1 (summarized in ref. [51]). In this compositional regime, all $Fe^{3+}$ is predicted to occupy the A-site, while $Al^{3+}$ fills the rest of the A-site and all of the

smaller B-site(e.g., refs. [52,53]) and therefore no spin transition of $Fe^{3+}$ is expected to take place in the B-site of Bdg in a pyrolitic lower mantle. Some recent experimental studies suggest that cation exchange between A-site $Fe^{3+}$ and B-site $Al^{3+}$ becomes more favorable at high $P–T$ conditions, driven by the volume collapse across the spin transition of the B-site $Fe^{3+}$ (ref. [22,40,51,54]). On the other hand, site exchange is not supported by theoretical calculations, which predict very limited migration of A-site $Fe^{3+}$ to the B-site (<~4%) throughout the lower mantle $P–T$ conditions (54,55). These studies and a recent experimental study on single-crystal Bdg[55] suggest that $Fe^{3+}$ in the B-site of Fe, Al-bearing Bdg is metastable and therefore most Bdg in Earth's mantle may contain no $Fe^{3+}$ in the B-site. Even in the absence of Fe–Al site exchange, however, multiple scenarios could give rise to domains in the mantle where the experimentally observed $Fe^{3+}$ spin transition occurs in Bdg. First, in Al, Si-poor, oxidized lithology, $Fe^{3+}$ may be forced into the Bdg B-site. For example, subducted harzburgite is depleted in Al with Al/Fe as low as 0.18[56]. If there is not enough Al + Si to fill the Bdg B-site, $Fe^{3+}$ may be driven by crystal chemistry to adopt this site[57]. Moreover, $Fe^{3+}$-rich materials, such as banded iron formation and goethite, could also be carried to the lower mantle by subducted slabs and would provide local chemical heterogeneous regions enriched in $Fe^{3+}$, with a high $Fe^{3+}$/Al ratio. Second, $Fe^{3+}$ may take the B-site of Bdg as a result of metastable arrangement of Fe during fast crystallization of melts in partially molten (hot and/or hydrous) regions. While spin transition in Bdg likely occurs in regions with either subducted $Fe^{3+}$-rich, Al-poor lithologies, or fast/metastable crystallization, our results demonstrate that a spin transition in these regions would not have a major effect on seismic velocities or electrical conductivity, but could influence other geophysical or geochemical processes.

Spin transitions have been suggested to weaken the lower mantle phase assemblage[32,33,58], offering a potential explanation for a viscosity minimum around 1600–2500 km depth inferred by geoid inversion studies[9,10], which may affect dynamics of subducted slabs and hot upwelling plumes[59,60]. However, studies of effects of spin transitions on deformation of lower mantle minerals have been limited to Fp[32,33,58]. Fp likely comprises <20% of the lower mantle phase assemblage and will only have a significant effect on viscosity if grains are interconnected. If the lower mantle is enriched in Si and adopts equilibrium texture[61,62], Bdg is the interconnected phase that will control deformation. Due to the high strength of Bdg relative to Fp[63], the viscosity of a dominantly Bdg lower mantle is high. Based on our experimental observations, the spin transition in $Fe^{3+}$-dominant Bdg occurs at similar depths and induces comparable reduction in both bulk modulus and bulk velocity as Fp (Supplementary Fig. 7). As a result, the spin transition in Bdg may also cause a comparable change in viscosity[32,33]. The decrease in viscosity during the spin transition and increase at higher pressures matches the observed broad valley in lower mantle viscosity profile with the minimum at about 1600–2500 km[9,10]. Together with the notion that the lower mantle may be more enriched in Bdg than previous estimation[18,61,62], the spin transition in $Fe^{3+}$-bearing Bdg thus may play an important role in controlling lower mantle dynamics.

With this new robust constraint on the EoS of $Fe^{3+}$-bearing Bdg, we can conclude that redox effects on bulk modulus and density of Bdg for normal mantle compositions are not detectable in the deep mantle by current geophysical methods (Fig. 4). The difference between physical properties of Bdg with HS $Fe^{2+}$, HS $Fe^{3+}$, LS $Fe^{3+}$, or even mixed spin $Fe^{3+}$ at lower mantle conditions is too small to be resolved by seismology. Along the lower mantle geotherm, the pressure range of the spin transition of the B-site $Fe^{3+}$ in Bdg is broadened by about 30 GPa[25,26], meaning

that a mixture of HS and LS B-site $Fe^{3+}$ in Bdg would coexist over ~800 km depth range. Although the mixed spin state of the $(Mg_{0.46}Fe^{3+}_{0.53})(Si_{0.49}Fe^{3+}_{0.51})O_3$ Bdg in this study at 300 K causes decrease of the bulk modulus (52%) and bulk sound speed (31%) (Supplementary Fig. 7), the temperature-induced broadening and lower $Fe^{3+}$-content in lower mantle Bdg will together decrease the magnitudes of the softening by ~100 times for lower mantle compositions and temperatures[25,26]. The mixed spin state in ferric but not ferrous Bdg provides the strongest signal for potentially observing contrast in $V_B$ between oxidized and reduced Bdg. If seismic tomography techniques improve precision in resolution of $V_B$ to 0.5%, valence states of iron in mantle Bdg could be resolved; for sensitivity to spin state, a precision closer to 0.01% would be required beneath about 1850 km (Fig. 4). For Mg# = Mg/(Mg + Fe) = 90 Bdg representative of the mantle, differences in oxidation state of iron result in a density difference up to ~0.3% (Fig. 4), far less than the 1.5–2% redox-induced density contrast required to rapidly separate oxidized materials from reduced materials in the early history of the Earth[20]. Moreover, the spin-transition-induced density increase makes the density contrast of Bdg with different $Fe^{3+}/\Sigma Fe$ ratios sharply fade away below the mid-mantle depth (Fig. 4 and Supplementary Fig. 5). Recent experimental and theoretical studies show that the $Fe^{3+}/\Sigma Fe$ ratio of Bdg is not constant but varies significantly across the lower mantle $P–T$ conditions[47,48]. Given the smooth density and sound velocity profiles of the lower mantle[31], the minor influence of both spin and valence states of iron in Bdg on its elastic properties may reconcile geophysical observations and mineral physics. Since both spin and valence states of iron in Bdg are invisible to seismic tomography, other mechanisms are required to explain observed lower mantle heterogeneities, such as a combination of regional enrichment in iron and deficiency in silicon[17,62].

## Methods

**Bdg synthesis.** Samples were synthesized from a mixture of approximately 1:1:1 molar ratios high purity (>99.99%) $Fe_2O_3$, MgO, and $SiO_2$ at 24 GPa and 1873 K for about 9 h using the multi-anvil apparatus at the University of Michigan. The resulting akimotoite was quenched from high temperature and slowly decompressed. $^{57}$Fe-enriched akimotoite was synthesized by the same method using $^{57}Fe_2O_3$ ($^{57}$Fe 94.3%) instead. The average composition of the recovered magnesium silicate samples is $Mg_{0.46(2)}Fe_{1.04(1)}Si_{0.49(1)}O_3$, based on electron microprobe analysis (SX-100; focused beam; accelerating voltage of 15 keV and beam current of 10 nA; forsterite ($Mg_2SiO_4$) was used as Mg and Si standard, while magnetite ($Fe_3O_4$) was used as Fe standard. Minor amounts of $Mg_{1.2(1)}Fe_{3.8(1)}O_7$ were found along grain boundaries of Bdg sample (Supplementary Fig. 1) and this phase may adopt the same structure as recently reported $Fe_5O_7$[64]. The ambient XRD pattern of $Mg_{0.46(2)}Fe_{1.04(1)}Si_{0.49(1)}O_3$ sample matches $R\overline{3}$ ilmenite structure with no contamination from the minor $Mg_{1.2(1)}Fe_{3.8(1)}O_7$ phase. The ambient unit cell volume of our akimotoite sample is 282.8 $Å^3$, consistent with 50% linear mixing between reported volumes for the isostructural $R\overline{3}$ end members $Fe_2O_3$[64] and $MgSiO_3$[65]. $Mg_{0.46(2)} Fe_{1.04(1)}Si_{0.49(1)}O_3$ akimotoite transforms to Bdg at ~24 GPa and 300 K and is fully recovered to ilmenite structure with the same lattice parameters as the initial values after decompression. As a result, the composition of the Bdg phase should be the same as akimotoite and the stoichiometric chemical formula of our Bdg sample is written as $(Mg_{0.46}Fe_{0.53})(Si_{0.49}Fe_{0.51})O_3$.

**DAC experiments.** Akimotoite samples were prepared for high-pressure experiments in symmetric-type DACs with pairs of 300-μm, 200-μm flat diamonds for pressure ranges up to 65.9 and 84.9 GPa, respectively. The sample chambers were confined by rhenium gaskets for XRD and hybrid-mode time-domain SMS measurements, while an X-ray transparent beryllium gasket was used for XES measurements. The gaskets were preindented to ~30 μm and then sample chambers with diameters approximately halves of the culet sizes were machined using the laser drilling system at HPCAT (Sector 16) of the Advanced Photon Source (APS), Argonne National Laboratory (ANL). About $20 \times 20 \times 7 μm^3$ polycrystalline akimotoite aggregates were loaded into the sample chambers. For XRD measurements, Au powder was spread on top of akimotoite samples to serve as pressure standard with minimal pressure gradient between samples and Au[36]. During XES and SMS measurements, pressures were determined from the edge of the diamond Raman peak recorded from the tip of the diamond anvil at the sample position before and after each data collection[66]. For XRD experiments, the COMPRES/GSECARS gas-

loading system at APS, ANL was used to load neon into the sample chamber as a hydrostatic pressure medium. For XES and SMS measurements, the pressure medium was silicone oil.

**XRD**. Angle-dispersive XRD measurements were performed at beamline 13-BM-C of the APS, ANL. The incident X-ray beam had a monochromatic wavelength of 0.434 Å and was focused to a spot size with a full width at half maximum of $15 \times 15 \, \mu m^2$. Diffracted X-rays were recorded on a MAR165 CCD detector. The sample-to-detector distance and the tilt angle and rotation angle of the image plate relative to the incident X-ray beam were calibrated by 1 bar diffraction of $LaB_6$. At intervals of 1–2 GPa, XRD images of the samples were recorded for an exposure time of 60 s. The XRD images were integrated using the software DIOPTAS. Diffraction patterns were analyzed using the software FullProf to examine the crystal structure and extract lattice parameters.

The compression curve of our Bdg sample exhibits softening between 43.1 and 52.5 GPa. In this pressure range a discontinuity is also observed in the corresponding normalized stress $F = P/3 f(1 + 2f)^{5/2}$ vs. Eulerian strain $f = [(V/V_0)^{-2/3} - 1]/2$ plot (Fig. 2). The horizontal segments below and above 43.1–52.5 GPa in F–f plot demonstrate that second order Birch–Murnaghan EoS is sufficient to fit the compression data (Fig. 2). The fraction of the HS state ($n_{LS}$) in the softening segment of the compression curve is determined by the method introduced by ref. [32]: $V = (1 - n_{LS})V_{HS} + n_{LS}V_{LS}$, and the corresponding bulk modulus ($K$) of the mixed spin state is calculated by the following equation:

$$\frac{V}{K} = (1 - n_{LS})\frac{V_{HS}}{K_{HS}} + n_{LS}\frac{V_{LS}}{K_{LS}} - (V_{LS} - V_{HS})\left(\frac{\partial n_{LS}}{\partial P}\right)_T, \qquad (1)$$

where $V_{HS}$ and $V_{LS}$ are the unit cell volume of HS and LS states at a given pressure $P$, respectively. The fitted HS fraction $n_{HS} = 1 - n_{LS}$ is shown in Fig. 3b and the calculated bulk modulus ($K$) and bulk sound velocity ($V_B$) are plotted against pressure in Supplementary Fig. 7.

**XES**. XES measurements were performed at beamline 16-ID-D of the APS and ANL at pressures up to 84.9 GPa at 300 K (Fig. 3). The incident X-ray beam with $5 \times 7 \, \mu m^2$ full width at half maximum was focused on the sample. Fluorescence signal was observed through the Be gasket. The incident X-ray energy was 11.3 keV with a bandwidth of ~1 eV. Fe $K_\beta$ emission was selected by silicon analyzer and reflected to a silicon detector with an energy step of about 0.3 eV[67]. Each spectrum took about 40 min and 1–3 spectra were taken to accumulate at least 30,000 counts at the Fe $K_\beta$ main peak at each pressure.

Each spectrum is composed of an Fe $K_\beta$ main peak and a well-resolved lower energy satellite $K_{\beta'}$ peak. Both integrated absolute difference (IAD) and integrated relative difference (IRD) methods[68] were used to quantitatively analyze the total spin moment. Spectra were first normalized to area and aligned to the position of the Fe $K_\beta$ main peak (Fig. 3a). Intensity difference between the sample and standards was integrated over the whole energy range (7018.3–7083.8 eV) for IAD, but only around the satellite $K_{\beta'}$ peak (7018.3–7054.0 eV) for IRD. Both the spectra of $Fe_2O_3$ and the sample at 1 bar served as HS references and $FeS_2$ at 1 bar was used as the LS reference. The spectra of references were collected using the same setup to prevent systematic error. The use of different HS standards generates <5% difference, which provides an estimate of uncertainty (Fig. 3b). The pressure range of the spin transition observed in XES is broader than that derived from softening of the compression curve (perhaps due to use of a less hydrostatic pressure medium in this experiment), but centered at the same average transition pressure of 48–49 GPa (Fig. 3b).

**Nuclear forward scattering**. Time-domain SMS measurements were performed at 26–71 GPa and 300 K at beamline 3ID-B of the APS. The storage ring was operated in hybrid mode, offering a ~50% longer time window than the typical 24-bunch mode for data collection and thus stronger constraints on the hyperfine parameters. The X-ray beam was focused to ~20 × 20 μm. Spectra were typically collected for 12 h. All SMS spectra were fitted using the CONUSS package using a two-site model with fixed equal intensity weighting based on the chemical formula (Supplementary Fig. 2). The small QS values of both sites relative to HS $Fe^{2+}$ and small difference in CS ($\Delta CS < 0.3 \, mm \, s^{-1}$) between these two sites demonstrate that all Fe in our Bdg sample is $Fe^{3+}$[69]. Because QS and CS values for Fe generally increase with increasing coordination[69], the site with smaller CS is assigned to the sixfold-coordinated B-site and the site with larger CS is assigned to the 8–12-fold-coordinated A-site. Across the spin transition at 43–53 GPa, QS of the A-site $Fe^{3+}$ increases by 0.1–0.2 mm s$^{-1}$, while that of the B-site $Fe^{3+}$ increases by 0.2–0.3 mm s$^{-1}$ (Supplementary Fig. 2). This moderate increase in QS across the spin transition of $Fe^{3+}$ is consistent with previous experimental studies on bridgmanite[34,35] (Supplementary Fig. 3). In comparison, only the lower bound of theoretically predicted QS of B-site LS $Fe^{3+}$ is marginally consistent with our results (Supplementary Fig. 3). Because QS of different sites and valence states can be similar, interpreting time-domain SMS data for Bdg requires long-time-window spectra for unique fits, clear evidence of spin transition in complementary XRD and XES results, and well-defined Bdg samples without alteration in compositions and oxidation state during high-pressure experiments.

**Electrical resistance measurements**. In situ high-pressure electric resistance was measured by a four-point-probe system at High Pressure Synergetic Consortium (HPSynC) at the APS. The resistance measurement system is composed of a Keithley 6221 current source, a 2182 A nanovoltmeter, and a 7001 voltage/current switch system. $Mg_{0.46}Fe_{1.04}Si_{0.49}O_3$ akimotoite sample was loaded into a symmetric DAC with 300-μm diamonds. A stainless steel gasket was first preindented to 15 GPa with 50 μm in thickness, then the indent was milled out and replaced by cubic boron nitride (cBN). Four 10-μm Au leads were pressed into contact with the sample and insulated from the stainless steel gasket by cBN powder (Supplementary Fig. 6). Current was supplied through two adjacent Au leads while the other two leads measured the corresponding voltage (marked in Supplementary Fig. 6). The first set of resistance measurements was collected during compression, then the pressure was released and the DAC was compressed again for the second set of resistance measurements (Fig. 5). The electrical conductivity was calculated by using the measured resistance, the distances between leads and established sample thickness before compression and after decompression. Due to its incompressibility, the thickness of cBN insert only changed by <10% between 20 GPa and up to 60 GPa, as observed in a test experiment. As a result, the uncertainty of calculated electrical conductivity caused by the sample dimension is likely to be <10%, which is supported by the reproducibility of the electrical conductivity derived from two successive runs in the same DAC (Fig. 5 and Supplementary Table 4).

**Data availability**. The datasets generated during and/or analyzed during the current study are available as Supplementary Information and from the corresponding authors.

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

## Acknowledgements

The authors thank Jung-Fu Lin, Renata Wentzcovitch, Allen McNamara, Jeroen Ritsema, Zhixue Du, Wenzhong Wang, Shuai Zhang, and Xiang Wu for helpful discussion. This work was supported by new faculty startup funding to S. M. Dorfman from Michigan State University and National Science Foundation EAR-1664332. Jie Li acknowledges support from NASA NNX15AG54G and NSF AST 1344133. Portions of the experiments were performed at GSECARS (sector 13), HPCAT (sector 16) and sector 3, Advanced Photon Source (APS), Argonne National Laboratory. GeoSoilEnviroCARS is supported by the National Science Foundation—Earth Sciences (EAR-1634415) and Department of Energy—GeoSciences (DE-FG02-94ER14466). Use of the COMPRES-GSECARS gas-loading system and the PX2 program were supported by COMPRES under NSF Cooperative Agreement EAR-1606856 and by GSECARS through NSF grant EAR-1634415 and DOE grant DE-FG02-94ER14466. HPCAT operations are supported by DOE-NNSA under Award No. DE-NA0001974, with partial instrumentation funding by NSF. Y.X. acknowledges the support of DOE-BES/DMSE under Award DE-FG02-

99ER45775. The Advanced Photon Source is a U.S. Department of Energy (DOE) Office of Science User Facility operated for the DOE Office of Science by Argonne National Laboratory under Contract No. DE-AC02-06CH11357.

## Author contributions

J.L. and S.M.D. designed research; J.L., S.M.D., F.Z., J.L., Y.W., D.Z., Y.X., W.B., and E.E. A performed research. J.L. and S.M.D. analyzed data; J.L. and S.M.D. wrote the paper.

## Additional information

**Competing interests:** The authors declare no competing interests.

