## [Peer Review File(PDF 428 kb) · Nature Communications]

Reviewers' comments:

Reviewer #1 (Remarks to the Author):

I thank the editor for giving me the opportunity to review the manuscript titled "Can valence and spin states be detected in Earth's lower mantle" by Liu and coauthors. This work helps us resolve the long-standing issues about spin state crossover in Fe³⁺-bearing bridgmanite and its possible consequences. Although, most of the conclusions about the spin crossover in Fe³⁺-bearing bridgmanite and its consequences for Earth's lower mantle were already known from previous computational studies, but convincing experimental support was very much needed. It is good that authors could prepare bridgmanite sample, which has just ferric iron. This success has led to provide convincing support for spin crossover.

I suggest this study may be published in nature communication after discussing the following points:

- 1) Work by Xu et al. 2015 (Xu, S., Shim, S.-H., Morgan, D., 2015. Origin of Fe³⁺ in Fe-containing, Al-free mantle silicate perovskite. EPSL.) should be discussed.
- 2) In line 75-76 and in 145-146, it is more likely that differences in spin crossover pressure in previous experimental studies are more likely due to different experimental protocol. It does not seem to depend on composition (i.e., iron concentration) significantly.
- 3) In line 104-106, the values of quadrupole splitting for Fe³⁺ is discussed. These measured values should also be compared (at least in supplementary document) with previous measurements (i.e., Lin and coauthors).
- 4) In line 116 and in 136, previous measurement by Mao et al. (GRL 2015) also observed the volume collapse due to Fe³⁺-spin crossover. Of course, that study had very less amount of ferric iron, but it may be good to compare amount of volume collapse per unit Fe³⁺ concentration. This may give some idea whether the presence of Fe²⁺ will have any effect on the equation of state or not.
- 5) In Figure 3b, it is being said that the high spin fractions (red open circles) are obtained using equation of state. May be some details should be discussed in supplementary section.
- 6) It is not very clear what authors want to convey in the sentence between line 234-236.
- 7) I am not sure if it is typo in line 241 (... , which may leave both elastic and elastic signatures...) or authors want to say something. It should be clarified.

Reviewer #2 (Remarks to the Author):

Review of Article "Can valence and spin states of iron be detected in Earth's lower mantle?" by Jiachao Liu, Susannah M. Dorfman, Feng Zhu, Jie Li, Yonggang Wang, Dongzhou Zhang, Yuming Xiao, Wenli Bi, E. Ercan Alp [Manuscript # NCOMMS-17-18486]

This manuscript investigates the effects of ferric iron Fe³⁺ on the density, compressibility and electrical conductivity of the most abundant mineral on Earth, bridgmanite (Mg, Fe, Al, Si)O₃. The hopes are that the effects of Fe³⁺ may provide observables for the heterogeneity found in Earth's mantle. Additionally, although they find that the spin transition of Fe³⁺ in the B site of bridgmanite is very clear and does indeed soften the bulk modulus a lot (~50%), once the high-temperatures and likely mantle compositions are taken in to account, the effects of Fe³⁺ on seismic heterogeneity will be minor.

I am impressed with this study and its thoroughness in the many methods used to explore the effects of Fe³⁺ in bridgmanite: XRD, XES, electrical conductivity and SMS. Together, the results from each of these experiments makes for a cohesive story. My own bias would be that Fe³⁺ would have a larger effect than it appears to have on the mantle. Oh well!

Overall, I like this paper and look forward to its publication.

A few comments:

Lines 41-44: This is an awkward sentence. Can you combine the two "anomaly" and refer to them as "anomalies"?

Figure 1: Please remove the background from the XRD pattern. It would make the presentation of your data more palatable.

Line 155: Add an "s" to "DAC"

Line 241: "elastic and elastic properties"? I think there is a typo here.

Lines 245-248: Awkward sentence. It's unclear what exactly you're trying to say here.

Line 275: How did you get 0.3% density difference for an Fe#10 bridgmanite? What Fe³⁺ content did you assume? Also, I don't think you define Fe#. Why not use Mg#?

Line 402: How much did the sample thickness change? How did you measure this? Did you take this into account when computing the conductivity (or resistivity)? What are your uncertainties? There is a paper on measuring conductivity in a DAC and how to take in to account the changing sample thickness that may be helpful (Montgomery et al., JOURNAL OF APPLIED PHYSICS 110, 043725 (2011)).

Table S4: "Sigma" symbols have been lost.

Reviewer #3 (Remarks to the Author):

This paper, "Can valence and spin states of iron be detected in Earth's lower mantle" by Liu et al, reports the results from an experimental study on the spin-transition of Al-free bridgmanite at 300 K and pressures consistent with lower mantle conditions. The system studied, $(\text{Mg}_{0.5}\text{Fe}_{0.5})(\text{Fe}_{0.5}\text{Si}_{0.5})\text{O}_3$, contains much higher iron content than bridgmanite crystallized from both peridotitic and basaltic lithologies, but the detailed and careful comparison of their results from different experimental techniques of such an "extreme case", enables the authors to interpolate their results to that of a pyrolitic mantle composition. This, in turn, allows the authors to uniquely resolve the elastic properties where previous studies have been hampered by large uncertainties, probably due to small changes in seismic velocities following a possible spin-transition in bridgmanite (br). This study clearly demonstrates that Al-free br undergoes a spin transition at 43-53 GPa, and that the high-to-low spin transition takes place at the crystallographic B site in the perovskite structure only. Based on these findings, they argue that spin-transitions and redox heterogeneities in the lower mantle has a minor effect on seismic wave-velocities in comparison to major cation concentration. This is supported by previously reported conductivity profiles from geomagnetic modeling data, and their own conductivity measurement.

In my opinion, there is no doubt that the authors have carried out a careful comparison of results from different experimental techniques, including complimentary electrical conductivity measurements, but my main concern with this study is that the use of Al-free samples do not represent realistic lower mantle lithologies. The discussion whether or not a spin-transition and valence heterogeneities can be captured seismically based on this model system, is therefore not relevant.

The Al/Fe(total) ratios of br in both basaltic and pyrolitic lithologies are larger than 1 (there is more Al than Fe(total) in bridgmanite): 1. Kesson, S.E., Fitz Gerald J.D., Shelley J.M., 1994. Mineral chemistry and density of subducted basaltic crust at lower-mantle pressures. *Nature* 372, 767–769. 2. Kesson, S.E., Fitz Gerald J.D., Shelley J.M., 1998. Mineralogy and dynamics of a pyrolite lower mantle. *Nature* 393, 252–255. 3. Hirose, K., Fei, Y., Ma, Y., Mao, H.K., 1999. The fate of subducted basaltic crust in the Earth's lower mantle. *Nature* 397, 53–56. 4. Hirose, K., 2002. Phase transitions in pyrolitic mantle around 670-km depth: implications for upwelling of plumes from the lower mantle. *J. Geophys. Res.* 107, B42078. 5. Hirose, K., Fei, Y., 2002. Subsolvus and melting phase relations of basaltic composition in the uppermost lower mantle. *Geochim. Cosmochim. Acta* 66, 2099–2108. 6. Hirose, K., Takafuji, N., Sata, N., Ohishi, Y. 2005. Phase transition and density of subducted MORB crust in the lower mantle. *Earth Planet. Sci. Lett.* 237, 239–251. 7. Ono, S., Ito, E., Katsura, T., 2001. Mineralogy of subducted basaltic crust (MORB) from 25 to 37 GPa, and chemical heterogeneity of the lower mantle. *Earth Planet. Sci. Lett.* 190, 57–63. 8. Trønnes, R.G., Frost, D.J., 2002. Peridotite melting and mineral-melt partitioning of major and minor elements at 22–24.5 GPa. *Earth Planet. Sci. Lett.* 197, 117–131. 9. Frost D.J., Liebske C., Langenhorst F., McCammon C.A., Trønnes R.G., Rubie D.C., 2004. Experimental evidence for the existence of iron-rich metal in the Earth's lower mantle. *Nature* 428, 409–412. 10. Liebske, C., Corgne, A., Frost, D.J., Rubie, D.C., Wood, B.J., 2005. Compositional effects on element partitioning between Mg-silicate perovskite and silicate melts. *Contrib. Mineral. Petrol.* 149, 113–128. 11. Murakami, M., K. Hirose, N. Sata, and Y. Ohishi (2005), Post-perovskite phase transition and mineral chemistry in the pyrolitic lowermost mantle, *Geophys. Res. Lett.*, 32, L03304, doi:10.1029/2004GL021956. 12. Ricolleau, A., Perrillat, J.-P., Fiquet, G., Daniel, I., Matas, J., Addad, A., Menguy, N., Cardon, H., Mezouar, M., Guignat, N., 2010. Phase relations and equation of state of a natural MORB: Implications for the density profile of subducted oceanic crust in the Earth's lower mantle. *J. Geophys. Res.* 115, B08202. 13. Irifune, T., Shinmei, T., MvCammon, C., Miyajima, N., Rubie, D.C., Frost, D.J., 2010. Iron partitioning and density changes of pyrolite in Earth's lower mantle. *Science* 327, 193–195. 14. Sinmyo, R., Hirose, K., Muto, S., Ohishi, Y., Yasuhara, A., 2011. The valence state and partitioning of iron in the Earth's lowermost mantle. *J. Geophys. Res.* 116, B07205. 15. Sinmyo, R., Hirose, K., 2013. Iron partitioning in pyrolitic lower mantle. *Phys. Chem. Mineral.* 40, 107–113. 16. Pradhan, G.K., Fiquet, G., Siebert, J., Auzende, A.-L., Morard, G., Antonangeli, D., Garbarino, G., 2015. Melting of MORB at core-mantle boundary. *Earth Planet. Sci. Lett.* 431, 247–251. Lin et al High-spin Fe²⁺ and Fe³⁺ in single-crystal aluminous bridgmanite in the lower mantle, *Geophys. Res. Lett.*, 43, 6952–6959 (2016).

Thus, when the mole fraction of Fe(total) is less than the mole fraction of Al, almost all of the iron atoms sit at the large A site in the perovskite structure, and very few can be found at the B site. This is also discussed by the authors in the introduction, further stated on L149 "Well characterised ..." and also discussed in the following papers: 1. Potapkin, V., McCammon, C., Glazyrin, K., Kantor, A., Kuzenko, I., Prescher, C., Sinmyo, R., Smirnov, G.V., Chumakov, A.I., Ruffer, R., Dubrovinsky, L. (2013) Effect of iron oxidation state on the electrical conductivity of the Earth's lower mantle. *Nature Comm.* 4, 1427. 2. Vanpeteghem, C.B., Angel, R.J., Ross, N.L., Jacobsen, S.D., Dobson, D., Litasov, K.D., Ohtani, E., 2006. Al, Fe substitution in the MgSiO₃ perovskite structure: a single-crystal X-ray diffraction study. *Phys. Earth Planet. Inter.* 155, 96–103. 3. Hsu, H., Yu, Y. G., Wentzcovitch, R. M., 2012. Spin crossover of iron in aluminous MgSiO₃ perovskite and post-perovskite. *Earth. Planet. Sci. Lett.* 645 359–360, 34–39.

Since iron undergoes a spin-transition when located at the B site only, as clearly demonstrated by the authors, there is very little evidence for a spin-transition of (ferric) iron and aluminum bearing br at lower mantle conditions in br crystallized from both pyrolitic and basaltic lithologies. (It should be noted that some studies has suggested that iron and aluminum can exchange positions assisted by a spin-transition: Fe(A) + Al(B) < Fe(B) + Al(A), but since the B site is thermally unavailable to ferric iron, as is shown by the energetics of fig 2a in: Hsu et al, "Spin crossover of iron in aluminous MgSiO₃ perovskite and post-perovskite" *Earth. Plan. Sci. Lett.* 459, 34–39 (2012) and explained by Mohn and Trønnes in "Iron spin state and site distribution in FeAlO₃-

bearing bridgmanite", *Earth. Plan. Sci. Lett.*, 440, 178-186 (2016), there will be no ferric iron at the B site in Al-bearing br and hence no spin transition will take place in oxidized br even at the lowermost mantle condition.

The question whether or not a spin-transition of iron can be detected seismically in the lower mantle using an Al-free model system, can therefore not be answered by the authors since the system investigated is not a realistic representation of the main lower mantle lithologies.

The second important question raised by the authors in the title of the paper is if seismology can distinguish between oxidized (ferric-iron rich br) and reduced (ferrous-iron rich br). Here, the authors compare molar volumes between relevant compounds - but there is apparently very little discussion about other factors that may influence the seismic wave velocities. That is Fe²⁺ and Fe³⁺ may affect bulk and shear moduli very differently! Anyway, my main worry is that the molar volume of Al-free br (investigated by the authors) is different from that of a realistic lower mantle br, and so this question can not be answered using Al-free samples either.

In addition, the explanation of why the molar volumes of the (fig 3 in suppl. mat.) based on comparison of Shannon's radii, is not very convincing. The authors show that HS ferric-bearing (Al-poor) br possess slightly larger molar volume than ferrous-bearing br (with similar iron content) before a spin-transition takes place, and similar volumes following a spin transition. Since both HS and LS Fe³⁺ are smaller than Fe²⁺, one could expect that the molar volume of ferrous-iron containing br is larger than that of ferric iron containing br. The authors explain the "large" volume of their Al-free Fe³⁺ bearing br compared to (Fe₂₊_{0.64},Fe₃₊_{0.24})SiO₃ by the strain introduced in the B sub-lattice due to the size-mismatch between HS-Fe³⁺ and Si⁴⁺ at the B site. It seems to me that the authors suggest that such a strained configuration possesses much larger volume compared to less strained configurations. However, such strain will, in part, be relieved by ordering the B-site cations in a rock-salt type pattern (see e.g; G. King and P. M. Woodward, "Cation order in perovskites" *J. Mater. Chem.*, 20 5785-5796 (2009)). In my opinion their argument of why the unit cell volume of their compound is so "large" is not very convincing, and the influence of the different charge-states of iron on bulk and shear moduli should be addressed.

Overall, I think this is an excellent experimental study, but the discussion is of little relevance in order to understand the influence of spin-transition and charge heterogeneities on seismic velocities, and therefore I can not recommend this paper for publication in nature communications in its current state.

Responses to editor comments and reviews
(Our replies are in blue and the revisions are red)

Reviewer #1 (Remarks to the Author):

I thank the editor for giving me the opportunity to review the manuscript titled “Can valence and spin states be detected in Earth’s lower mantle” by Liu and coauthors. This work helps us resolve the long-standing issues about spin state crossover in Fe³⁺-bearing bridgmanite and its possible consequences. Although, most of the conclusions about the spin crossover in Fe³⁺-bearing bridgmanite and its consequences for Earth’s lower mantle were already known from previous computational studies, but convincing experimental support was very much needed. It is good that authors could prepare bridgmanite sample, which has just ferric iron. This success has led to provide convincing support for spin crossover.

I suggest this study may be published in nature communication after discussing the following points:

1) Work by Xu et al. 2015 (Xu, S., Shim, S.-H., Morgan, D., 2015. Origin of Fe³⁺ in Fe-containing, Al-free mantle silicate perovskite. EPSL.) should be discussed.

Cited Xu et al. 2015 in line 257-261 and 307-311 as ref. 57.

‘In addition, subducted harzburgite is depleted in Al with Al/Fe as low as 0.18 (ref. 56) and represents a relatively oxidized environment. This low Al-content will enhance Fe³⁺ concentration in the Bdg B-site and thus enhance the importance of the spin transition in Bdg in regions rich in subducted lithosphere⁵⁷.’

‘Recent experimental and theoretical studies show that the Fe³⁺/ΣFe ratio of Bdg is not constant but varies significantly across the lower mantle *P-T* conditions^{49,50,57,58}. Given the smooth density, sound velocity profiles of the lower mantle³¹, the minor influence of both spin and valence states of iron in Bdg on its elastic properties may reconcile geophysical observations and mineral physics.’

2) In line 75-76 and in 145-146, it is more likely that differences in spin crossover pressure in previous experimental studies are more likely due to different experimental protocol. It does not seem to depend on composition (i.e., iron concentration) significantly.

We agree, and note in lines 149-152: ‘Differences between observed spin transition pressures are unlikely to be explained by compositional differences alone as had been suggested by computational work²⁶: our sample exhibits a spin transition pressure in between reported pressures in previous experiments on Bdg but has the highest Fe³⁺ content.’

3) In line 104-106, the values of quadrupole splitting for Fe³⁺ is discussed. These measured values should also be compared (at least in supplementary document) with previous measurements (i.e., Lin and coauthors).

Values for QS of Fe³⁺ in previous studies based on different methods exhibit a wide range, and some of the differences are likely due to non-uniqueness of fitting nuclear resonant scattering data. We chose to focus here on observations using energy-domain Mossbauer because interpretation of these data is relatively transparent. Our model is also supported by studies of other compounds with Fe³⁺ in 6-fold coordinated sites (e.g. Pasternak et al. 2002). A more thorough comparison is provided in *Methods - Nuclear forward scattering* and in *supplementary Fig.6*.

In addition, we modified the relevant sentence in the main text, line 108-110: ‘These values are consistent with the hyperfine parameters for Fe³⁺ of Bdg derived from synchrotron-based energy-domain Mössbauer spectroscopy^{34,35} (Supplementary Fig. 6).’

4) In line 116 and in 136, previous measurement by Mao et al. (GRL 2015) also observed the volume collapse due to Fe³⁺-spin crossover. Of course, that study had very less amount of ferric iron, but it may be good to compare amount of volume collapse per unit Fe³⁺ concentration. This may give some idea whether the presence of Fe²⁺ will have any effect on the equation of state or not.

We add new statement about Mao et al. 2015 in line139-141: ‘In comparison, Ref. 24 reported a 0.5% reduction in unit cell volume at 18-25 GPa with 0.02 Fe³⁺ per formula unit (pfu), which is higher but comparable with theoretical prediction²⁶.’

5) In Figure 3b, it is being said that the high spin fractions (red open circles) are obtained using equation of state. May be some details should be discussed in supplementary section.

We elaborate this in line 370-383: ‘The compression curve of our Bdg sample exhibits softening between 43.1 – 52.5 GPa. In this pressure range a discontinuity is also observed in the corresponding finite Eulerian strain $F = P/3f(1+2f)^{5/2}$ vs. normalized stress $f = [(V/V_0)^{-2/3}-1]/2$ plot (Fig. 2). The horizontal segments below and above 43.1 – 52.5 GPa in F-f plot demonstrate that 2nd order Birch-Murnaghan equations of state (BM-EoS) is sufficient to fit the compression data (Fig. 2). The fraction of the high-spin state (n_{LS}) in the softening segment of the compression curve is determined by the method introduced by ref. 32: $V = (1-n_{LS})V_{HS} + n_{LS}V_{LS}$, and the corresponding bulk modulus (K) of the mixed spin state is calculated by the following equation:

$$\frac{V}{K} = (1 - n_{LS}) \frac{V_{HS}}{K_{HS}} + n_{LS} \frac{V_{LS}}{K_{LS}} - (V_{LS} - V_{HS}) \left(\frac{\partial n_{LS}}{\partial P} \right)_T$$

Where V_{HS} and V_{LS} are the unit cell volume of HS and LS states at a given pressure P , respectively. The fitted high spin fraction $n_{HS}=1- n_{LS}$ is shown in Fig. 3b and the calculated bulk modulus (K) and bulk sound velocity (V_B) is plotted against pressure in supplementary Fig. 2a.’

6) It is not very clear what authors want to convey in the sentence between line 234-236.

We rewrite this part in line 247 –261: ‘Whether Fe³⁺ enters the B-site of Bdg through this coupled-substitution mechanism and further undergoes the spin transition in the lower mantle depends on both the Al/Fe ratio of Bdg and *P-T* conditions. For Bdg samples synthesized from pyrolitic starting materials representing a lower mantle lithology, observed Al/Fe³⁺ ratios are consistently greater than 1 (summarized in ref. 53). In this compositional regime, all Fe³⁺ is predicted to occupy the A-site while Al³⁺ fills the rest of the A-site and all of the smaller B-site (e.g. ref. 54,55). However, experimental studies have observed site exchange between A-site Fe³⁺ and B-site Al³⁺ at high *P-T* conditions, which is suggested to be driven by the volume collapse across the spin transition of the B-site Fe³⁺ (ref. 22,41,53). This site exchange suggests the spin transition will be observed in the mantle even in Al-rich lithologies. In addition, subducted harzburgite is depleted in Al with Al/Fe as low as 0.18 (ref. 56) and represents a relatively oxidized environment. This low Al-content will enhance Fe³⁺ concentration in the Bdg B-site and thus enhance the importance of the spin transition in Bdg in regions rich in subducted lithosphere⁵⁷.’

7) I am not sure if it is typo in line 241 (... , which may leave both elastic and elastic signatures...) or authors want to say something. It should be clarified.

Change in line 265-266 to ‘which may leave signatures on both thermoelastic and transport properties in the midmantle^{53,58}.’

Reviewer #2 (Remarks to the Author):

Review of Article “Can valence and spin states of iron be detected in Earth’s lower mantle?” by Jiachao Liu, Susannah M. Dorfman, Feng Zhu, Jie Li, Yonggang Wang, Dongzhou Zhang, Yuming Xiao, Wenli Bi, E. Ercan Alp [Manuscript # NCOMMS-17-18486]

This manuscript investigates the effects of ferric iron Fe³⁺ on the density, compressibility and electrical conductivity of the most abundant mineral on Earth, bridgmanite (Mg, Fe, Al, Si)O₃. The hopes are that the effects of Fe³⁺ may provide observables for the heterogeneity found in Earth’s mantle. Additionally, although they find that the spin transition of Fe³⁺ in the B site of bridgmanite is very clear and does indeed soften the bulk modulus a lot (~50%), once the high-temperatures and likely mantle compositions are taken in to account, the effects of Fe³⁺ on seismic heterogeneity will be minor.

I am impressed with this study and its thoroughness in the many methods used to explore the effects of Fe³⁺ in bridgmanite: XRD, XES, electrical conductivity and SMS. Together, the results from each of these experiments makes for a cohesive story. My own bias would be that Fe³⁺ would have a larger effect than it appears to have on the mantle. Oh well!

Thanks to Reviewer 2 for your support!

Overall, I like this paper and look forward to its publication.

A few comments:

Lines 41-44: This is an awkward sentence. Can you combine the two “anomaly” and refer to them as “anomalies”?

Figure 1: Please remove the background from the XRD pattern. It would make the presentation of your data more palatable.

Done.

Line 155: Add an “s” to “DAC”

Done.

Line 241: “elastic and elastic properties”? I think there is a typo here.

Change in line 265-266 to ‘which may leave signatures on both thermoelastic and transport properties in the midmantle^{52,57}’.

Lines 245-248: Awkward sentence. It’s unclear what exactly you’re trying to say here.

Change in line 270-276: ‘However, studies of effects of spin transitions on deformation of lower mantle minerals have been limited to Fp^{32,33,59}. Fp likely comprises <20% of the lower mantle phase assemblage and will only have a significant effect on viscosity if grains are interconnected. If the lower mantle is enriched in Si and adopts equilibrium texture^{62,63}, Bdg is the interconnected phase that will control deformation. Due to the high strength of Bdg relative to Fp⁶⁴, the viscosity of a dominantly-Bdg lower mantle is high.’

Line 275: How did you get 0.3% density difference for an Fe#10 bridgmanite? What Fe³⁺ content did you assume? Also, I don’t think you define Fe#. Why not use Mg#?

Change to ‘Mg#=Mg/(Mg+Fe)=90’ in line 301.

0.3 % density difference is between pure Fe²⁺-Bdg and Fe³⁺-Bdg, which is based on the linear fit for density-Fe# relation in Fig. 4a.

Line 402: How much did the sample thickness change? How did you measure this? Did you take this into account when computing the conductivity (or resistivity)? What are your uncertainties? There is a paper on measuring conductivity in a DAC and how to take in to account the changing sample thickness that may be helpful (Montgomery et al., JOURNAL OF APPLIED PHYSICS 110, 043725 (2011)).

Detailed information is added in the caption of Fig. 5: ‘The uncertainty is smaller than the symbol size.’;

Line 437-439: ‘A stainless steel gasket was first pre-indented to 15 GPa with 50 μm in thickness, then the indent was milled out and replaced by cubic boron nitride (cBN).’;

Line 447-451: ‘Due to its incompressibility, the thickness of cBN insert only changed by less than 10% between 20 GPa up to 60 GPa, as observed in a test experiment. As a result, the uncertainty of calculated electrical conductivity caused by the sample dimension is likely to be less than 10%, which is supported by the reproducibility of the electrical conductivity derived from two successive runs in the same DAC (Fig. 5).’

Table S4: “Sigma” symbols have been lost.
Added.

Reviewer #3 (Remarks to the Author):

This paper, "Can valence and spin states of iron be detected in Earth's lower mantle" by Liu et al, reports the results from an experimental study on the spin-transition of Al-free bridgmanite at 300 K and pressures consistent with lower mantle conditions. The system studied, $(\text{Mg}_{0.5}\text{Fe}_{0.5})(\text{Fe}_{0.5}\text{Si}_{0.5})\text{O}_3$, contains much higher iron content than bridgmanite crystallized from both peridotitic and basaltic lithologies, but the detailed and careful comparison of their results from different experimental techniques of such an "extreme case", enables the authors to interpolate their results to that of a pyrolitic mantle composition. This, in turn, allows the authors to uniquely resolve the elastic properties where previous studies have been hampered by large uncertainties, probably due to small changes in seismic velocities following a possible spin-transition in bridgmanite (br). This study clearly demonstrate that Al-free br undergoes a spin transition at 43-53 GPa, and that the high-to-low spin transition takes place at the crystallographic B site in the perovskite structure only. Based on these findings, they argue that spin-transitions and redox heterogeneities in the lower mantle has a minor effect on seismic wave-velocities in comparison to major cation concentration. This is supported by previously reported conductivity profiles from geomagnetic modeling data, and their own conductivity measurement.

In my opinion, there is no doubt that the authors have carried out a careful comparison of results from different experimental techniques, including complimentary electrical conductivity measurements, but my main concern with this study is that the use of Al-free samples do not represent realistic lower mantle lithologies. The discussion whether or not a spin-transition and valence heterogeneities can be captured seismically based on this model system, is therefore not relevant.

First of all, we would like to thank Reviewer 3 for testifying to the quality of our experimental study, and for devoting substantial time and effort to this review. We appreciate this list of relevant references and questions which have helped us to clarify our manuscript. We respectfully disagree with the assertion that Al-free samples cannot contribute to the discussion of the spin transition in the lower mantle, for the reasons summarized above in our letter and

detailed below. In short, although our composition (and temperature conditions) do not match the mantle, these experiments clarify the crystal chemistry of this system in a way that would be challenging or impossible with Al-bearing samples, and the results provide a strong bound on the maximum effect of spin and valence state on observable properties.

The Al/Fe(total) ratios of br in both basaltic and pyrolitic lithologies are larger than 1 (there is more Al than Fe(total) in bridgmanite): 1. Kesson, S.E., Fitz Gerald J.D., Shelley J.M., 1994. Mineral chemistry and density of subducted basaltic crust at lower-mantle pressures. *Nature* 372, 767–769. 2. Kesson, S.E., Fitz Gerald J.D., Shelley J.M., 1998. Mineralogy and dynamics of a pyrolite lower mantle. *Nature* 393, 252–255. 3. Hirose, K., Fei, Y., Ma, Y., Mao, H.K., 1999. The fate of subducted basaltic crust in the Earth's lower mantle. *Nature* 397, 53–56. 4. Hirose, K., 2002. Phase transitions in pyrolitic mantle around 670-km depth: implications for upwelling of plumes from the lower mantle. *J. Geophys. Res.* 107, B42078. 5. Hirose, K., Fei, Y., 2002. Subsolvus and melting phase relations of basaltic composition in the uppermost lower mantle. *Geochim. Cosmochim. Acta* 66, 2099–2108. 6. Hirose, K., Takafuji, N., Sata, N., Ohishi, Y. 2005. Phase transition and density of subducted MORB crust in the lower mantle. *Earth Planet. Sci. Lett.* 237, 239–251. 7. Ono, S., Ito, E., Katsura, T., 2001. Mineralogy of subducted basaltic crust (MORB) from 25 to 37 GPa, and chemical heterogeneity of the lower mantle. *Earth Planet. Sci. Lett.* 190, 57–63. 8. Trønnes, R.G., Frost, D.J., 2002. Peridotite melting and mineral-melt partitioning of major and minor elements at 22–24.5 GPa. *Earth Planet. Sci. Lett.* 197, 117–131. 9. Frost D.J., Liebske C., Langenhorst F., McCammon C.A., Trønnes R.G., Rubie D.C., 2004. Experimental evidence for the existence of iron-rich metal in the Earth's lower mantle. *Nature* 428, 409–12. 10. Liebske, C., Corgne, A., Frost, D.J., Rubie, D.C., Wood, B.J., 2005. Compositional effects on element partitioning between Mg-silicate perovskite and silicate melts. *Contrib. Mineral. Petrol.* 149, 113–128. 11. Murakami, M., K. Hirose, N. Sata, and Y. Ohishi (2005), Post-perovskite phase transition and mineral chemistry in the pyrolitic lowermost mantle, *Geophys. Res. Lett.*, 32, L03304, doi:10.1029/2004GL021956. 12. Ricolleau, A, Perrillat, J.-P., Fiquet, G., Daniel, I., Matas, J., Addad, A., Menguy, N., Cardon, H., Mezouar, M., Guignat, N., 2010. Phase relations and equation of state of a natural MORB: Implications for the density profile of subducted oceanic crust in the Earth's lower mantle. *J. Geophys. Res.* 115, B08202. 13. Irifune, T., Shinmei, T., MvCammon, C., Miyajima, N., Rubie, D.C., Frost, D.J., 2010. Iron partitioning and density changes of pyrolite in Earth's lower mantle. *Science* 327, 193–195. 14. Sinmyo, R., Hirose, K., Muto, S., Ohishi, Y., Yasuhara, A., 2011. The valence state and partitioning of iron in the Earth's lowermost mantle. *J. Geophys. Res.* 116, B07205. 15. Sinmyo, R., Hirose, K., 2013. Iron partitioning in pyrolitic lower mantle. *Phys. Chem. Mineral.* 40, 107–113. 16. Pradhan, G.K., Fiquet, G., Siebert, J., Auzende, A.-L., Morard, G., Antonangeli, D., Garbarino, G., 2015. Melting of MORB at core-mantle boundary. *Earth Planet. Sci. Lett.* 431, 247–251. Lin et al High-spin Fe²⁺ and Fe³⁺ in single-crystal aluminous bridgmanite in the lower mantle, *Geophys. Res. Lett.*, 43, 6952–6959 (2016).

Thus, when the mole fraction of Fe(total) is less than the mole fraction of Al, almost all of the iron atoms sit at the large A site in the perovskite structure, and very few can be found at the B site. This is also discussed by the authors in the introduction, further stated on L149 "Well characterised ..." and also discussed in the following papers: 1. Potapkin, V., McCammon, C., Glazyrin, K., Kantor, A., Kuppenko, I., Prescher, C., Sinmyo, R., Smirnov, G.V., Chumakov, A.I., Ruffer, R., Dubrovinsky, L. (2013) Effect of iron oxidation state on the electrical conductivity of

the Earth's lower mantle. *Nature Comm.* 4, 1427. 2. Vanpeteghem, C.B., Angel, R.J., Ross, N.L., Jacobsen, S.D., Dobson, D., Litasov, K.D., Ohtani, E., 2006. Al, Fe substitution in the MgSiO₃ perovskite structure: a single-crystal X-ray diffraction study. *Phys. Earth Planet. Inter.* 155, 96–103. 3. Hsu, H., Yu, Y. G., Wentzcovitch, R. M., 2012. Spin crossover of iron in aluminous MgSiO₃ perovskite and post-perovskite. *Earth. Planet. Sci. Lett.* 645 359-360, 34–39.

Since iron undergoes a spin-transition when located at the B site only, as clearly demonstrated by the authors, there is very little evidence for a spin-transition of (ferric) iron and aluminum bearing br at lower mantle conditions in br crystallized from both pyrolytic and basaltic lithologies. (It should be noted that some studies has suggested that iron and aluminum can exchange positions assisted by a spin-transition: Fe(A) + Al(B) < Fe(B) + Al(A), but since the B site is thermally unavailable to ferric iron, as is shown by the energetics of fig 2a in: Hsu et al, "Spin crossover of iron in aluminous MgSiO₃ perovskite and post-perovskite" *Earth. Plan. Sci. Lett.*, 459, 34-39 (2012) and explained by Mohn and Trønnes in "Iron spin state and site distribution in FeAlO₃-bearing bridgmanite", *Earth. Plan. Sci. Lett.*, 440, 178-186 (2016), there will be no ferric iron at the B site in Al-bearing br and hence no spin transition will take place in oxidized br even at the lowermost mantle condition.

To address this important point, we have added discussion and reference to the review of this body of work in Shim et al. *PNAS* 2017, which supports the spin transition even in Al-bearing lithologies due to site exchange. In addition, we note that Al-poor lithologies are also important to the mantle, e.g. the estimated Al/Fe ratio of subducted harzburgite is only 0.18 (Xu et al., *EPSL*). Geodynamic modelling shows that mantle convection could concentrate harzburgite composition locally in the lower mantle (e.g. Ballmer et al., 2015, *Sci. Adv.*, Xu et al., 2008, *EPSL*). In other words, the lower mantle is very likely to be chemically heterogeneous (reviewed by Garnero et al., 2016, *Nature Geoscience*), and the harzburgitic heterogeneities could be very Al-poor and therefore Bdg formed there would experience spin transition for B-site Fe³⁺. Therefore, the spin transition could be important regardless of Al-content, and Al-poor Bdg is still relevant to the lower mantle.

New discussion in line 247-261: 'Whether Fe³⁺ enters the B-site of Bdg through this coupled-substitution mechanism and further undergoes the spin transition in the lower mantle depends on both the Al/Fe ratio of Bdg and *P-T* conditions. For Bdg samples synthesized from pyrolytic starting materials representing a lower mantle lithology, observed Al/Fe³⁺ ratios are consistently greater than 1 (summarized in ref. 53). In this compositional regime, all Fe³⁺ is predicted to occupy the A-site while Al³⁺ fills the rest of the A-site and all of the smaller B-site (e.g. ref. 54,55). However, experimental studies have observed site exchange between A-site Fe³⁺ and B-site Al³⁺ at high *P-T* conditions, which is suggested to be driven by the volume collapse across the spin transition of the B-site Fe³⁺ (ref. 22,41,53). This site exchange suggests the spin transition will be observed in the mantle even in Al-rich lithologies. In addition, subducted harzburgite is depleted in Al with Al/Fe as low as 0.18 (ref. 56) and represents a relatively oxidized environment. This low Al-content will enhance Fe³⁺ concentration in the Bdg B-site and thus enhance the importance of the spin transition in Bdg in regions rich in subducted lithosphere⁵⁷.'

The question whether or not a spin-transition of iron can be detected seismically in the lower mantle using an Al-free model system, can therefore not be answered by the authors since the system investigated is not a realistic representation of the main lower mantle lithologies.

See our summary above in the body of our response letter. We agree with the reviewer that this would be a concern if our conclusion was opposite: that the spin transition and valence state have important effects on geophysical observables. However, our results provide an upper bound on the effect of the spin transition/valence state, and find it insignificant and undetectable. An Al-bearing lithology would further weaken the effect of spin transition of Fe^{3+} in the geophysical properties of Bdg. A parallel set of experiments for Al-bearing Bdg would be highly unlikely to change the conclusions of this work. Moreover, experiments on Al-bearing Bdg would produce ambiguous results that could be questioned on the basis of what the site occupancy of Al is during the experiments and whether it matches that in the real mantle. The approach we have chosen yields a clear constraint on effects of Fe^{3+} in both sites in Bdg. Our conclusion that **the effect of the spin transition/valence state on observable physical properties of the lower mantle is undetectably small** is an important contribution to a broader discussion in the geophysics and geochemistry communities. It allows us to simplify our interpretation of seismology.

The second important question raised by the authors in the title of the paper is if seismology can distinguish between oxidized (ferric-iron rich br) and reduced (ferrous-iron rich br). Here, the authors compare molar volumes between relevant compounds - but there is apparently very little discussion about other factors that may influence the seismic wave velocities. That is Fe^{2+} and Fe^{3+} may affect bulk and shear moduli very differently! Anyway, my main worry is that the molar volume of Al-free br (investigated by the authors) is different from that of a realistic lower mantle br, and so this question cannot be answered using Al-free samples either.

The influence of Fe^{2+} and Fe^{3+} on bulk modulus is illustrated in Fig. 4 and discussed in the paragraph at line 167-194.

The influence of Al on the elastic properties is discussed in line 196-206: 'For a given concentration of Fe, the presence of Al in Bdg has been observed to have relatively minor effects on density and bulk modulus^{19,25} (Fig. 4) and may suppress the spin transition by occupying the B-site (see *Implications* below). As a result, experiments on Fe,Al-bearing compositions have been unable to unambiguously determine whether and under what conditions spin transitions take place in the mantle. The effects of spin and valence states of Fe on density and bulk compressibility are expected to be even less significant in Al-bearing lithologies in the mantle. Although shear properties cannot be constrained by our experimental data, theoretical calculations have predicted that the effects of trivalent cations and/or spin transition of the B-site Fe^{3+} on shear modulus are even smaller than on bulk modulus²⁵. Therefore, the incorporation of trivalent cations in Bdg is not expected to cause obvious elastic anomalies in the lower mantle.'

In addition, the explanation of why the molar volumes of the (fig 3 in suppl. mat.) based on comparison of Shannons radii, is not very convincing. The authors show that HS ferric-bearing (Al-poor) br possess slightly larger molar volume than ferrous-bearing br (with similar iron content) before a spin-transition takes place, and similar volumes following a spin transition.

Since both HS and LS Fe³⁺ are smaller than Fe²⁺, one could expect that the molar volume of ferrous-iron containing br is larger than that of ferric iron containing br. The authors explain the "large" volume of their Al-free Fe³⁺ bearing br compared to (Fe²⁺_{0.64}Fe³⁺_{0.24})SiO₃ by the strain introduced in the B sub-lattice due to the size-mismatch between HS-Fe³⁺ and Si⁴⁺ at the B site. It seems to me that the authors suggest that such a strained configuration possesses much larger volume compared to less strained configurations. However, such strain will, in part, be relieved by ordering the B-site cations in a rock-salt type pattern (see e.g; G. King and P. M. Woodward, "Cation order in perovskites" J. Mater. Chem, 20 5785-5796 (2009)). In my opinion their argument of why the unit cell volume of their compound is so "large" is not very convincing, and the influence of the different charge-states of iron on bulk and shear moduli should be addressed.

Supplementary Fig.3 clearly shows that our Fe³⁺-only Bdg has the highest unit cell volume below the spin transition pressures among all existing experimental data. It is simply because our sample has the highest Fe content among all existing bdg sample and does not have significant amounts of vacancies. Although (Fe²⁺_{0.64}Fe³⁺_{0.24})SiO₃ from Ismailova et al. (2016) has significant amount of large Fe²⁺, it also has 0.12 A-site vacancy PFU, making its volume smaller than ours below ~45 GPa. The relevant statement is in line 170-174: 'With the highest Fe-content among synthesized Bdg, our Fe³⁺-only Bdg has the largest unit cell below the pressure of the spin transition. Above the spin transition pressure of B-site Fe³⁺, the unit cell volume of our Fe³⁺-Bdg collapses to match volumes of Fe²⁺-dominant Bdg with similar total Fe-content (Supplementary Fig. 3).'

Reviewer 3 is correct that 1-bar radii only relate qualitatively to this discussion. To simplify the discussion and avoid confusion on a point that may only interest specialist readers, we remove the inset for atomic radii in supplementary Fig. 3 and delete the text about 1-bar radii comparison between different cations.

The comparison of elastic properties of Fe²⁺, HS Fe³⁺, LS Fe³⁺-Bdg is discussed in line 167-194 and illustrated in Fig. 4 and supplementary Fig. 3. 'For iron-rich compositions, the elastic properties and spin-transition-induced softening in Fe³⁺-Bdg can be easily distinguished from elastic properties of Fe²⁺-dominant Bdg, but for mantle-relevant amounts of iron this difference becomes insignificant (Fig. 4). With the highest Fe-content among synthesized Bdg, our Fe³⁺-only Bdg has the largest unit cell observed to date for Bdg below the pressure of the spin transition. Above the spin transition pressure of B-site Fe³⁺, the unit cell volume of our Fe³⁺-Bdg collapses to match volumes of Fe²⁺-dominant Bdg with similar total Fe-content (Supplementary Fig. 3). Consequently, redox heterogeneity cannot be determined from density heterogeneity once the spin-transition of B-site Fe³⁺ is complete in the deep lower mantle (Fig. 4a and Supplementary Fig. 4). The bulk moduli K of both HS and LS Fe³⁺-rich Bdg are lower than that of Fe²⁺-dominant Bdg (Fig. 4b). At a representative mid-lower-mantle pressure of 80 GPa (corresponding to a depth of 1850 km), K of HS (Mg_{0.46}Fe³⁺_{0.53})(Si_{0.49}Fe³⁺_{0.51})O₃ Bdg is 9.3% lower than the extrapolated K for FeSiO₃ Bdg, and K of (Mg_{0.46}Fe³⁺_{0.53})(Si_{0.49}Fe³⁺_{0.51})O₃ Bdg with B-site LS Fe³⁺ is 11.1% lower than that of FeSiO₃ Bdg (Fig. 4b). The magnitudes of these differences in K are comparable to softening caused by A-site vacancy⁴². The corresponding bulk sound velocity for Fe³⁺-dominant Bdg exhibits a similar trend as bulk modulus (Fig. 4c). The heterogeneity parameter $\partial \ln V_B / \partial X_{Fe}$ for Fe³⁺-Bdg is 0.15; this is 1.5x of the 0.1 obtained for Fe²⁺-

dominant Bdg¹⁹, resulting in a stronger velocity anomaly for an oxidized mantle heterogeneity. If interpolated to a typical mantle composition with iron content $2\text{Fe}/(\text{Mg}+\text{Fe}+\text{Al}+\text{Si}) \sim 0.1$ in Bdg (ref. 43) at pressure corresponding to the mid lower mantle, differences in density, bulk modulus, and bulk sound velocity between reduced and oxidized bridgmanite at 80 GPa are up to 0.3%, 1.1% and 0.5%, respectively. These small differences have been within experimental uncertainties for studies with less Fe, but can be resolved by our study of well-characterized Fe-rich Bdg samples with careful high-pressure experimental design. Given the fact that lower-mantle temperatures would reduce the difference in density and sound velocity between Fe²⁺- and Fe³⁺-bearing bdg, reduced and oxidized Bdg with mantle-relevant iron-content will exhibit almost identical seismic velocities in the deep lower mantle.?’

Reviewers' comments:

Reviewer #3 (Remarks to the Author):

Dear Editor,

I have now read the responses to my comments and I'm still not convinced by the arguments that Al free br represents a good model to guide seismologists. However, I do agree that this compound can add some constraints on the influence of the spin-transition of ferric iron on observable physical properties. I strongly recommend that the authors change part of the discussion with an appropriate reference to some recent literature (along the lines suggested below) before I can be recommended this paper for publication in nature communications.

On p. 7 in their responses to one of my comments the authors reply: "To address this important point still relevant to the lower mantle"

First of all, I think we all agree that the a spin-transition of ferric iron in Al-bearing br takes place at the B-site only.

To support the *presence* of ferric iron at the B-site in br crystallized from different possible lower mantle assemblages, the authors are referring to results by Shim *et al* (PNAS 114, 6468, (2017)). This paper reports a drop in $Fe^{3+}/Fe(tot)$ content in a 35-60 GPa pressure range, and then an increase in $Fe^{3+}/Fe(tot)$ from 60 GPa to about 110 GPa. The increase in $Fe^{3+}/Fe(tot)$ within the lowermost mantle is explained by the stabilization of low spin Fe^{3+} at the B site due to the volume decrease following the exchange: $Fe^{3+}(A) + Al(B) \leftrightarrow Fe^{3+}(B) + Al(A)$.

However, there are also several experimental studies carried out on Al-bearing br that do *not* support such a spin-transition and where very little ferric iron is found at the B site even at lowermost mantle conditions. These papers include: Potapkin *et al* (Nat. Commun., 1427,4), Kuznetsov *et al* (EPSL, 423, 78-86) and an excellent recent single-crystal study on $Mg_{0.89}Fe_{0.12}Al_{0.11}Si_{0.89}O_3$ by Lin *et al.* (GRL, 43, 6952-6959). In the latter paper by Lin *et al* the authors write: "Therefore, even though the B-site Fe^{3+} in bridgmanite has been reported to transition to the low-spin state at high pressures [e.g., Catalli *et al.*, 2011; Hsu *et al.*, 2012; Lin *et al.*, 2012; Mao *et al.*, 2015; Muir and Brodholt, 2016], the negligible amount of the B-site Fe^{3+} in (Fe,Al)-bearing bridgmanite is unlikely to play a major role in influencing the lower mantle geophysics and geochemistry. These results also indicate that the dominant A-site Fe^{3+} does not migrate to the B site at relevant lower mantle conditions." This finding is strongly supported by DFT calculations which show that the exchange-mechanism is energetically unfavourable because the B-site is not thermally accessible to ferric iron (in any spin state!).

So why do some experiments support an occupation of the B-site whereas others do not ? and why do some experimental studies show an occupation of the B-site whereas computational studies clearly show that in equilibrated samples of br, the B site should not be occupied by iron even at lowermost mantle conditions ?

To address these questions, it is of interest to note that, most, if not all, studies that supports a spin transition (iron is at the B site) are prepared from either gel or glass, whereas studies that do not see a spin-transition (iron is at the A site only) are prepared by grinding the binary oxides components together. This is discussed by Mohn and Trønnes (EPSL, 440,179 (2016)) and they suggests that the use of glass/gel as starting material for synthesis can cause the iron to become kinetically trapped during sample preparations.

To conclude, the state-of-the-art of knowledge appears to support that in fully equilibrated samples of br, ferric iron will remain at the A site at all mantle conditions as long as the Si+Al content is larger than 1 (per MgSiO₃ f.u.). I therefore, don't think that the recent paper by Shim et al (PNAS; 114, 6468 (2017)) gives relevant support to the presence of ferric iron at the B site in br because the model used by the authors to explain the occupation of iron at the B site, is hampered by lack of thermodynamic support from ab initio DFT calculations.

Therefore: the new discussion L247-261 (highlighted in red at bottom of page 7 in the authors responses) should be updated along the lines sketched out above with an appropriate reference to the literature and in particular the authors should discuss what we have learnt from these DFT calculations. The sentence "However experimental studies have observed site exchange ... in the mantle even in Al-rich lithologies" should be removed, and the discussion and updated as lined out with a relevant list of references, where also studies that do not see such a spin-transition and recent comp. studies are appropriately discussed".

To motivate their choice of model the authors discuss subducted hartzburgite depleted in Al with Al/Fe ratio as low as 0.18. First of all, I do not understand what the authors mean by "represent a relatively oxidised environment" ? and why is this important to highlight ? I don't think there is any doubt that br crystallized from hartzburgite at lower mantle conditions also will contain ferric iron! The key point is that, as long as Si+Al is equal to or larger than 1 (per f.u. MgSiO₃), ferric iron will not enter the B site even with low Al/Fe(tot) ratio. However, if Si+Al is smaller than one, then ferric iron may be forced into the B site. There appear to be few experimental analysis of hartzburgite itself, but to address if ferric iron could enter the B site a meaningful comparison would possibly be San Carlos Olivine (where there is no Al at all). The analysis of the binary components from these compounds (e.i. Nomura et al (2011) and Auzende et al (2008)) suggests that there should be enough Si available to fill the B site in br in the MgO+MgSiO₃ assemblage.

Of course, one can not rule out that there might be domains in the lowermost mantle where ferric iron are either forced into the B site or are kinetically trapped at this site, when crystallized from, say, partial melts in hot regions, and so I agree that Al-free assemblages may contribute to put some constraints on the possibility to capture such effects seismically.

Overall, have no doubt in that this is an excellent experimental study and, in a way, I do agree that this compound can add some constraints on the influence of the spin-transition of ferric iron on observable physical properties. This warrants a publication in nature communication. However, still I can not recommend publication of this paper as long as the discussion is halted by the lack of convincing arguments of why we should study Al-free assemblages to guide seismologists.

Responses to reviewer's comments
(Our replies are in blue and the revisions are red)

On p. 7 in their responses to one of my comments the authors reply: "To address this important point still relevant to the lower mantle"

First of all, I think we all agree that the a spin-transition of ferric iron in Al-bearing br takes place at the B-site only.

To support the *presence* of ferric iron at the B-site in br crystallized from different possible lower mantle assemblages, the authors are referring to results by Shim *et al* (PNAS 114, 6468, (2017)). This paper reports a drop in $\text{Fe}^{3+}/\text{Fe}(\text{tot})$ content in a 35-60 GPa pressure range, and then an increase in $\text{Fe}^{3+}/\text{Fe}(\text{tot})$ from 60 GPa to about 110 GPa. The increase in $\text{Fe}^{3+}/\text{Fe}(\text{tot})$ within the lowermost mantle is explained by the stabilization of low spin Fe^{3+} at the B site due to the volume decrease following the exchange: $\text{Fe}^{3+}(\text{A}) + \text{Al}(\text{B}) \leftrightarrow \text{Fe}^{3+}(\text{B}) + \text{Al}(\text{A})$.

However, there are also several experimental studies carried out on Al-bearing br that do *not* support such a spin-transition and where very little ferric iron is found at the B site even at lowermost mantle conditions. These papers include: Potapkin *et al* (Nat. Commun., 1427,4), Kuppenko *et al* (EPSL, 423, 78-86) and an excellent recent single-crystal study on $\text{Mg}_{0.89}\text{Fe}_{0.12}\text{Al}_{0.11}\text{Si}_{0.89}\text{O}_3$ by Lin *et al.* (GRL, 43, 6952–6959). In the latter paper by Lin *et al* the authors write: "Therefore, even though the B-site Fe^{3+} in bridgmanite has been reported to transition to the low-spin state at high pressures [e.g., Catalli *et al.*, 2011; Hsu *et al.*, 2012; Lin *et al.*, 2012; Mao *et al.*, 2015; Muir and Brodholt, 2016], the negligible amount of the B-site Fe^{3+} in (Fe,Al)-bearing bridgmanite is unlikely to play a major role in influencing the lower mantle geophysics and geochemistry. These results also indicate that the dominant A-site Fe^{3+} does not migrate to the B site at relevant lower mantle conditions." This finding is strongly supported by DFT calculations which show that the exchange-mechanism is energetically unfavourable because the B-site is not thermally accessible to ferric iron (in any spin state!).

So why do some experiments support an occupation of the B-site whereas others do not ? and why do some experimental studies show an occupation of the B-site whereas computational studies clearly show that in equilibrated samples of br, the B site should not be occupied by iron even at lowermost mantle conditions ?

To address these questions, it is of interest to note that, most, if not all, studies that supports a spin transition (iron is at the B site) are prepared from either gel or glass, whereas studies that do not see a spin-transition (iron is at the A site only) are prepared by grounding the binary oxides components together. This is discussed by Mohn and Trønnes (EPSL, 440,179 (2016)) and they suggests that the use of glass/gel as starting material for synthesis can cause the iron to become kinetically trapped during sample preparations.

We fully agree with the reviewer that the existing experimental data for spin state of Fe in Bdg are controversial, especially those involving bridgmanite synthesis by laser heating technique in diamond anvil cells. After realizing this problem, some recent papers, such as Ballaran et al. (EPSL, 2014), Mao et al (GRL, 2015) and our study here use well-characterized pre-synthesized samples for the equation of state measurements. More detailed comments on this experimental issue can be found in line 58-63 and line 156-165.

To conclude, the state-of-the-art of knowledge appears to support that in fully equilibrated samples of br, ferric iron will remain at the A site at all mantle conditions as long as the Si+Al content is larger than 1 (per MgSiO₃ f.u.). I therefore, don't think that the recent paper by Shim et al (PNAS; 114, 6468 (2017)) gives relevant support to the presence of ferric iron at the B site in br because the model used by the authors to explain the occupation of iron at the B site, is hampered by lack of thermodynamic support from ab initio DFT calculations.

Therefore: the new discussion L247-261 (highlighted in red at bottom of page 7 in the authors response) should be updated along the lines sketched out above with an appropriate reference to the literature and in particular the authors should discuss what we have learnt from these DFT calculations. The sentence "However experimental studies have observed site exchange ... in the mantle even in Al-rich lithologies" should be removed, and the discussion and updated as lined out with a relevant list of references, where also studies that do *not* see such a spin-transition and recent comp. studies are appropriately discussed".

We modify the text according the reviewer's comments (Line 254-279): 'In this compositional regime, all Fe³⁺ is predicted to occupy the A-site while Al³⁺ fills the rest of the A-site and all of the smaller B-site (e.g. ref. 54,55) and therefore no spin transition of Fe³⁺ is expected to take place in the B-site of Bdg in a pyrolitic lower mantle. Some recent experimental studies suggest that cation exchange between A-site Fe³⁺ and B-site Al³⁺ becomes more favorable at high *P-T* conditions, driven by the volume collapse across the spin transition of the B-site Fe³⁺ (ref. 22,41,53,56). On the other hand, site exchange is not supported by theoretical calculations, which predict very limited migration of A-site Fe³⁺ to the B-site (<~4%) throughout the lower mantle *P-T* conditions (54,55). These studies and a recent experimental study on single-crystal Bdg (ref. 57) suggest that Fe³⁺ in the B-site of Fe,Al-bearing bridgmanite is metastable and therefore most Bdg in Earth's mantle may contain no Fe³⁺ in the B-site. Even in the absence of Fe-Al site exchange, however, multiple scenarios could give rise to domains in the mantle where the experimentally observed Fe³⁺ spin transition occurs in Bdg. First, in Al,Si-poor, oxidized lithology Fe³⁺ may be forced into the Bdg B-site. For example, subducted harzburgite is depleted in Al with Al/Fe as low as 0.18 (ref. 58). If there is not enough Al+Si to fill the Bdg B-site, Fe³⁺ may be driven by crystal chemistry to adopt this site. Moreover, Fe³⁺-rich materials such as banded iron formation (BIF) and goethite could also be carried to the lower mantle by subducted slabs and would provide local chemical heterogeneous regions enriched in Fe³⁺, with a high Fe³⁺/Al ratio. Second, Fe³⁺ may take the B site of Bdg as a result of metastable arrangement of Fe during fast crystallization of melts in partially-molten (hot and/or hydrated) regions. While spin transition in Bdg likely occur in regions with either subducted Fe³⁺-rich, Al-poor lithologies⁵⁹ or fast/metastable crystallization, our results demonstrate that a spin transition in

these regions would not have a major effect on seismic velocities or electrical conductivity, but could influence other geophysical or geochemical processes.'

To motivate their choice of model the authors discuss subducted harzburgite depleted in Al with Al/Fe ratio as low as 0.18. First of all, I do not understand what the authors mean by "represent a relatively oxidized environment" ? and why is this important to highlight ? I don't think there is any doubt that br crystallized from harzburgite at lower mantle conditions also will contain ferric iron! The key point is that, as long as Si+Al is equal to - or larger than 1 (per f.u. MgSiO₃), ferric iron will not enter the B site even with low Al/Fe(tot) ratio. However, if Si+Al is smaller than one, then ferric iron may be forced into the B site. There appear to be few experimental analysis of harzburgite itself, but to address if ferric iron could enter the B site a meaningful comparison would possibly be San Carlos Olivine (where there is no Al at all). The analysis of the binary components from these compounds (e.i. Nomura et al (2011) and Auzende et al (2008)) suggests that there should be enough Si available to fill the B site in br in the MgO+MgSiO₃ assemblage.

We deleted the statement 'represent a relatively oxidized environment'.

Harzburgite is conveyed to the lower mantle by cold subducted slabs, which isn't expected to be molten. Therefore, the partitioning study between silicate and melt by Nomura is not directly relevant. The composition of bridgmanite forming in harzburgitic lithology is expected to be controlled by the bulk composition, which has a Mg/(Si+Al) ratio as high as 1.52 (Xu et al., 2008, EPSL). This leaves sufficient room for Fe³⁺ to enter the B-site of bridgmanite and drives spin transition in the lower mantle.

Of course, one can not rule out that there might be domains in the lowermost mantle where ferric iron are either forced into the B site or are kinetically trapped at this site, when crystallized from, say, partial melts in hot regions, and so I agree that Al-free assemblages may contribute to put some constraints on the possibility to capture such effects seismically.

Overall, have no doubt in that this is an excellent experimental study and, in a way, I do agree that this compound can add some constraints on the influence of the spin-transition of ferric iron on observable physical properties. This warrants a publication in nature communication. However, still I can not recommend publication of this paper as long as the discussion is halted by the lack of convincing arguments of why we should study Al-free assemblages to guide seismologists.